

# Dynamic Processes Dominating Ozone Variability in Warm Seasons of 2014–2018 over the Yangtze River Delta Region, China

Da Gao[1], Min Xie[1*], Jane Liu[2,3], Tijian Wang[1], Chaoqun Ma[1,a], Haokun Bai[1], Xing Chen[1]

[1] School of Atmospheric Sciences, Joint Center for Atmospheric Radar Research of CMA/NJU, CMA-NJU Joint Laboratory for Climate Prediction Studies, Jiangsu Collaborative Innovation Center for Climate Change, Nanjing University, Nanjing 210023, China

[2] College of Geographic Sciences, Fujian Normal University, Fuzhou 350007, China

[3] Department of Geography and Planning, University of Toronto M5S 3G3, Canada

[a] now at: Minerva Research Group, Max Planck Institute for Chemistry, Mainz, Germany

------------------------------------------------------------------

* Corresponding author. School of Atmospheric Sciences, Nanjing University, Nanjing 210023, China. minxie@nju.edu.cn (M. Xie)

**Abstract**: Ozone ($O_3$) pollution is of great concern in the Yangtze River Delta (YRD) region of China, and the regional $O_3$ pollution is closely associated with dominant weather systems. With a focus on the warm seasons (April–September) from 2014 to 2018, we quantitatively analyze the characteristics of $O_3$ variations over the YRD, the impacts of large-scale and synoptic-scale circulations on the variations and the associated meteorological controlling factors, based on observed ground-level $O_3$ and meteorological data. Our analysis suggests an increasing trend of the regional mean $O_3$ concentration in the YRD at 1.81 ppb per year over 2014-2018. Spatially, the empirical orthogonal function (EOF) analysis suggests the dominant mode accounting for 65.70% variation in $O_3$, implying that an increase in $O_3$ is the dominant tendency in the entire YRD. Meteorology is estimated to increase the regional mean $O_3$ concentration by 2.81 ppb at most from 2014 to 2018. Relative humidity is found to be the most influential meteorological factor impacting $O_3$ concentration. As the atmospheric circulation can affect local meteorological factors and $O_3$ levels, we identify five dominant synoptic weather patterns (SWPs) in the warm seasons in the YRD using the t-mode principal component analysis (PTT) classification. The typical weather systems of SWPs include western Pacific Subtropical High (WPSH) under SWP1, a continental



high under SWP2, an extratropical cyclone under SWP3, a southern low pressure and WPSH
under SWP4 and the north China anticyclone under SWP5. The annual variations of all five SWPs
are favorable to the increase in $O_3$ concentrations over 2014-2018. Moreover, the change in SWP
intensity contributes more to the $O_3$ inter-annual variation than the SWP frequency change. The
SWP intensity change includes the weakening and northward-extending of the western Pacific
subtropical high (WPSH) under SWP1, the weakening of the continental high under SWP2, an
extratropical cyclone strengthening under SWP3, the southern low pressure weakening and WPSH
weakening under SWP4, and the north China anticyclone weakening under SWP5. All these
changes prevent the water vapor in the southern sea from being transported to the YRD, and
increase air temperature in the YRD. In addition, the descending motions strengthen in the YRD
located behind the trough and in front of the ridge due to the strengthening of the ridge and trough
in the westerlies. Then, the strengthened descending motion leads to less cloud cover and strong
solar radiation, which are favorable to $O_3$ formation and accumulation. Finally, we reconstruct an
EOF mode 1 time series that shows high correlation with the original $O_3$ time series, and the
reconstructed time series performs well in defining the change in SWP intensity according to the
unique feature under each of the SWPs.

## 1. Introduction

As an air pollutant, surface ozone ($O_3$) is harmful to human health and vegetation growth, such
as damaging human lungs (Jerrett et al. 2009; Day et al. 2017) and destroying forest and
agricultural crops (Yue et al. 2017). In recent years, after reducing the emissions following
"Thirteenth Five-Year Plan" Comprehensive Work Plan for Energy Saving and Emission
Reduction since 2016, concentrations of many pollutants have decreased over the past few years
in China, but not for $O_3$. Furthermore, heavy $O_3$ pollutions occur more frequently and more
severely in China than those in Japan, South Korea, Europe and the United States (Lu et al. 2018).
Li et al. (2018) proposed that the rapid decrease of fine particulate matter (PM) in China is a
reason for such $O_3$ increase by slowing down the aerosol sink of hydro-preoxy radicals. Yet, the
contribution of meteorological factors to the $O_3$ increase is unclear.
Surface $O_3$ is mainly formed through complex and nonlinear photochemical reactions of
volatile organic compounds (VOCs) and nitrogen oxides ($NO_x$) exposed to the sunlight. Ozone



formation is sensitive to concentrations of NO$_x$ and VOCs, i.e., O$_3$ formation can be NO$_x$-limited
or VOC-limited regimes depending on concentrations of NO$_x$ and VOCs (Xie et al. 2014; Jin and
Holloway 2015). Meteorology could also affect O$_3$ levels through modulation of photochemical
reactions, advection, convection and turbulent transport, as well as dry and wet depositions (Liu et
al. 2013). Synoptic weather patterns (SWPs) and the associated meteorological conditions can
impact long-term and daily O$_3$ variations. Understanding the mechanisms of meteorological
influences on O$_3$ variations and quantifying such influences would help provide effective
emission-controlling plans for O$_3$ pollution.
Severe O$_3$ pollution episodes are accompanied with specific local meteorological conditions,
such as high temperature, strong solar radiation, drying condition and stagnant weather (Jacob and
Winner 2009; Doherty et al. 2013; Pu et al. 2017; Zhang et al. 2018). Moreover, local
meteorological conditions are often related to specific synoptic-scale and large-scale atmospheric
circulation systems. For example, O$_3$ pollution in the eastern United States is notably influenced
by the cyclone frequency (Leibensperger et al. 2008), latitude of the polar jet over eastern North
America (Barnes and Fiore. 2013) and the behavior of the quasi-permanent Bermuda High (Fiore
et al. 2003, Wang et al. 2016). In China, Yang et al. (2014) illustrated that the changes in
meteorological parameters, associated with the East Asian summer monsoon, lead to 2–5 %
inter-annual variations in surface O$_3$ concentrations over the central-eastern China. Zhao and
Wang et al. (2017) found that a significantly strong western Pacific subtropical high (WPSH)
could result in higher relative humidity (RH), more clouds, more rainfall, and less ultraviolet
radiation, finally leading to less O$_3$ formation. Using model simulation, Shu et al. (2016)
investigated the synergistical impact of the WPSH and typhoon on O$_3$ level in Yangtze River Delta
region.
As known, a region is influenced by different weather systems. Weather classification, as a
way to distinguish the different large-scale and synoptic-scale atmospheric circulation systems, is
widely used in exploring connections between weather patterns and O$_3$ levels (Han et al. 2020;
Gao et al. 2020). Gao et al. (2020) discussed influences of SWPs on O$_3$ levels, and revealed
differences in O$_3$ pollution levels due to the minor changes in atmospheric circulations. However,
spatially, it is uncertain that how the change in SWPs could lead to O$_3$ pollution in detail,





especially in the YRD. For the northern China and the PRD region, Liu et al. (2019) quantified the
impact of synoptic circulation patterns on $O_3$ variability in the northern China from April to
October during 2013–2017. Yang et al. (2019) quantitatively assessed the impacts of
meteorological factors and the precursor emissions on the long-term trend of ambient $O_3$ over the
PRD region. Yet, whether variations in SWPs can lead to $O_3$ increases has not be sufficiently
addressed.
Due to the ever-growing $O_3$ level in the YRD (Tong et al. 2017; Gao et al. 2017), the studies
on characteristics of $O_3$ variation and the underlying mechanisms for the variation are urgently
required. To this end, here the $O_3$ variations in space and time, as well as 5-year trend, in the YRD
is quantitatively investigated, and the mechanisms of meteorological influences on the $O_3$
variations are analyzed. Especially, the characteristics of the corresponding SWPs are discussed in
detailed. The remainder of this paper is organized as follows. Data and methods are introduced in
section 2. The inter-annual variation and 5-year trend and spatial variation characteristics are
illustrated in section 3.1. The impact of meteorological factors on the $O_3$ variation is discussed in
section 3.2. The main SWPs and the effects of their change on the $O_3$ variation are described in
section 3.3. Section 3.4 discusses the contributions of the SWP intensity and frequency change to
the inter-annual variation and trend of $O_3$. Finally, the conclusion and discussions are shown in
section 4.

**2. Data and methods**
**2.1. $O_3$ and meteorological datasets**
The maximum daily 8-hours average $O_3$ data are available from the National Environmental
Monitoring Center of China, which were acquired from the air quality real-time publishing
platform (http://106.37.208.233:20035). The hourly observation data of meteorological factors
including air temperature (T), RH, wind speed (WS) and sunshine duration (SD) in the warm
seasons from April to September over 2014–2018 were acquired from the National Meteorological
Center of China Meteorological Administration (http://eng.nmc.cn). 26 cities are selected as
typical cities representative of the YRD according to the "Urban agglomeration on Yangtze River
Delta" approved by China's State Council in 2016. In this paper, the term "$O_3$ concentration"
refers to the maximum daily 8-hours average $O_3$ concentration unless stated otherwise.

**2.2. Linear trend analyses**

In order to characterize the $O_3$ variation in the warm seasons during 2014–2018 over the
YRD, a linear trend method based on monthly anomalies is used (see Equation 1), which has been
widely used to calculate the trends of time series with seasonal cycles and autocorrelation. The $O_3$
monthly anomalies are more precise than $O_3$ monthly means because of the reducing impact of
missing data. Using this method, Cooper et al. (2020) and Lu et al. (2020) quantified the $O_3$ trend
in 27 globally distributed remote locations and the whole China. In addition, anomalies of monthly
average $O_3$ concentration are defined as the difference between the individual monthly mean and
the monthly mean of 2014–2018. The parametric linear trend is calculated by using the
generalized least-squares method with auto-regression.
$$y_t = b + kt + \alpha \cos\left(\frac{2\pi M}{6}\right) + \beta \sin\left(\frac{2\pi M}{6}\right) + R_t \qquad (1),$$
where $y_t$ represents the monthly anomaly, $t$ is the monthly index from April to September
during 2014–2018, $b$ denotes the intercept, $k$ is the linear trend, $\alpha$ and $\beta$ are coefficients for a
6-month harmonic series (M ranges from 1 to 6) which is used to account for potentially
remaining seasonal signals, and $R_t$ represents a normal random error series.

**2.3. Meteorological adjustment**

The meteorological adjustment, a statistical method, is applied to quantify the impact of
meteorology on $O_3$ variation through removing such impact in the original $O_3$ data. It is similar to
a model simulation that keeps the emission levels fixed but allows meteorology to vary. Yet, this
method requires much less computing resources than a model simulation. The method is
introduced in detail as follows.
In the meteorological adjustment, the observed $O_3$ and meteorological data are separated into
long-term, seasonal, and short-term data (Rao and Zurbenko 1994a, b). The
Kolmogorov-Zurbenko (KZ) filter can be expressed as follows.
$$R(t) = L(t) + S(t) + W(t) \qquad (2),$$
where $R(t)$ represents the raw time series data, $L(t)$ the long-term trend on a timescale of years,
$S(t)$ the seasonal variation on a timescale of months, and $W(t)$ the short-term component on a





timescale of days.

In order to remove the high-pass signal, the KZ filter carries out $p$ times of iterations of a

moving average with the window length $m$, which is defined as
$Y_i = \frac{1}{m} \sum_{j=-k}^{k} R_{i+j}$      (3)
where $R$ is the original time series, $i$ an index for the time of iteration, $j$ an index for sampling
inside the window, and $k$ the number of sampling on one side of the window. The window length
$m = 2k + 1$. $Y$ is the input time series after one iteration. Different scales of motions are obtained by
changing the window length and the number of iterations (Milanchus et al. 1998; Eskridge et al.
1997). The filter periods of less than $N$ days can be calculated with window length $m$ and the
number of iteration $p$, as follows:
$m \times p^{\frac{1}{2}} \leq N$      (4).
Therefore, the cycles of 33 days can be removed by a KZ (15, 5) filter with the window length of
15 and 5 iterations. In the following equation 5, BL(t) is the $O_3$ and meteorological time series
obtained by KZ(15,5) filter and refers to their baseline variations which are the sum of the long
term L(t) and the seasonal component S(t)..
$BL(t) = KZ_{(15,5)} = L(t) + S(t) = KZ_{(183,3)} + S(t)$      (5).
The long-term trend is separated from the raw data obtained by KZ (183, 3) with the periods of >
632 days, and then the seasonal and the short-term component $W(t)$ can be defined as
$S(t) = KZ_{(15,5)} - KZ_{(183,3)}$      (6),
$W(t) = X(t) - BL(t) = X(t) - KZ_{(15,5)}$      (7).
After KZ filtering, the meteorological adjustment is conducted by the multivariate regression
between the $O_3$ concentration and meteorological factors such as T, RH, wind speed and sunshine
duration (Wise and Comrie 2005; Papanastasiou et al. 2012).
$A_{BL}(t) = a_{BL} + \sum b_{BLi} \cdot M_{BLi} + \epsilon_{BL}(t)$      (8),
$A_W(t) = a_W + \sum b_{Wi} \cdot M_{Wi} + \epsilon_W(t)$      (9),

$\epsilon(t) = \epsilon_{BL}(t) + \epsilon_W(t)$      (10),

$A_{ad}(t) = \epsilon(t) + \sum b_{BLi} \cdot \overline{M}_{BLi} + \sum b_{Wi} \cdot \overline{M}_{Wi} + a_{BL} + a_W$      (11).
the multivariate regression models between baseline and short-term $O_3$ and meteorological factors
are shown in equations 8 and 9. The $A_{BL}(t)$ and $M_{BLi}$ represent the sum of the long term L(t)



and the seasonal component S(t) of $O_3$ concentration and meteorological factors. The $A_W(t)$ and
$M_{Wi}$ represent the short-term W(t) of $O_3$ concentration and meteorological factors. The $a$ and $b$
are the fitted parameters, and $i$ is time point (days). $\epsilon(t)$ is the residual term. The average
meteorological condition $\overline{M}$ at the same calendar date during the 5 years is regarded as the base
condition for that date, and the meteorological adjustment is conducted against the base condition.
By these steps, $A_{ad}(t)$ refers to the meteorologically adjusted $O_3$ variation with the homogenized
annual variation in meteorological conditions. The difference between raw $O_3$ time series and
$A_{ad}(t)$ represents the meteorological impact.

**2.4. Classification of SWPs**

In order to find the detailed variation characteristics of SWPs, we first extract the

predominant SWPs in the warm seasons over the YRD using a weather classification method.
Common objective classification methods include using predefined type, the leader algorithm, the
cluster analysis, optimization algorithms and eigenvectors (Philipp et al. 2016). The PTT method,
a simplified variant of t-mode principal component analysis using orthogonal rotation, is used to
classify SWPs during 2014–2018. It is one of the methods for weather classification in European
Cooperation in Science and Technology Action 733 (Philipp et al. 2016), which is widely used in
atmospheric sciences (Hou et al. 2019).

**2.5. FNL and ERA-Interim meteorological data**

The National Center for Environmental Prediction Final Operational Global Analysis (FNL)

data (http://rda.ucar.edu/datasets/ds083.2/) produced by the Global Data Assimilation System are
used in classifying SWPs and analyzing atmospheric circulations. The data have a horizontal
resolution of 2.5 °×2.5 °, with 144×73 horizontal grids available every 6 hours. From the near
surface layer to 10 hPa, there are 17 pressure levels in the vertical direction. The data of the
geopotential height and wind at 500 hPa and 850 hPa, the vertical wind ($\Omega$), T and RH are used in
this study. At the same time, the total cloud cover (TCC) and solar radiation (SR) from
ERA-interim are supplemented in this study, which have the same temporal and spatial resolutions
as the FNL data.
The FNL geopotential height field at 850 hPa can capture the synoptic circulation variations
over the YRD well (Shu et al. 2017). In this study, we use the geopotential height at 850 hPa from
April to September during 2014–2018 as the input for the PTT.

**2.6. Reconstruction of O₃ concentration based on SWP**
To quantify the inter-annual variability captured by the variations (frequency and intensity) in
the synoptic weather patterns, Yaranl (1992) provided an algorithm to find the contribution of
SWP frequency variation to the inter-annual O₃ variation. The specific calculation is as follows.
$\overline{\overline{O_{3m}}}(fre) = \sum_{k=1}^{6} \overline{O_{3k}} F_{km}$          (12),
where $\overline{\overline{O_{3m}}}(fre)$ is the reconstructed mean O₃ concentration influenced by the frequency
variation in SWPs from April to September for year $m$, $\overline{O_{3k}}$ is the 5-year mean O₃ concentration
for SWP $k$, and $F_{km}$ is the occurrence frequency of SWP $k$ during April–September for year $m$.
Hegarty et al. (2007) suggested that changes in the SWP include both frequency change and
intensity change. The intensity of SWPs represents the location and strength of the weather system.
Moreover, they noted that the environmental and climate-related contributions to the inter-annual
variations of O₃ could be better separated by considering these two changes. So, Equation12 is
modified into the following form.
$\overline{\overline{O_{3m}}}(fre + int) = \sum_{k=1}^{6}(\overline{O_{3k}} + \Delta O_{3km}) F_{km}$          (13),
where $\overline{\overline{O_{3m}}}(fre + int)$ is the reconstructed average O₃ concentration influenced by the
frequency and intensity changes of SWPs from April to September for year $m$; $\Delta O_{3km}$ is the
modified difference on the fitting line, which is obtained through a linear fitting of the annual O₃
concentration anomalies ($\Delta O_3$) to the SWP intensity index (SWPII) for SWP $k$ in year $m$. $\Delta O_{3km}$
represents the part of the annual observed O₃ oscillation caused by the intensity variation in each
SWP. Hegarty et al. (2007) used the domain averaged sea level pressure to represent the
circulation intensity index (CII). Liu et al. (2019) reconstructed the inter-annual O₃ level in the
northern China using the center pressure of the lowest pressure system. But we find the intensity
variation in each SWP is different when O₃ increases. So we select different SWPII under each
pattern according to the characteristics of high O₃ concentration. Lastly, we select the maximum
height in zone-1 (25 °N–40 °N, 110 °E–130 °E), the maximum height in zone-2 (20 °N–50 °N,





90 E–140 E) and the mean height in zone-3 (10 N–40 N, 110 E–130 E). Detailed demonstration
is introduced in section 3.5.

**3. Results and discussion**
**3.1. Spatio-temporal variations of $O_3$ in the YRD region**
**3.1.1. Inter-annual variations of $O_3$**

Fig. 1a shows the time series of the anomalies of the monthly mean $O_3$ concentration over the

YRD from April to September during 2014–2018, as well as the corresponding linear fitting curve.
Figure 1b shows the annual variation in the total number of days with $O_3$ concentration exceeding
the national standard during the period. As shown in Fig. 1a, the monthly mean $O_3$ concentration
in the warm seasons increases over 2014-2018, reaching the maximum of 37.44 ppb in 2017 and
maintaining at a high level in 2018. Specifically, $O_3$ concentration in the YRD shows a large
increasing trend of 1.81 ppb (5.21%) per year, which is slightly higher than that in the entire China
(5.00% per year, Lu et al. 2020). Meanwhile, the annual average days with $O_3$ exceeding the
standard also show an increasing trend, reaching a peak in 2017 and maintaining at a high level in
2018. In all, both means and extremes of $O_3$ concentration have increased over the YRD.

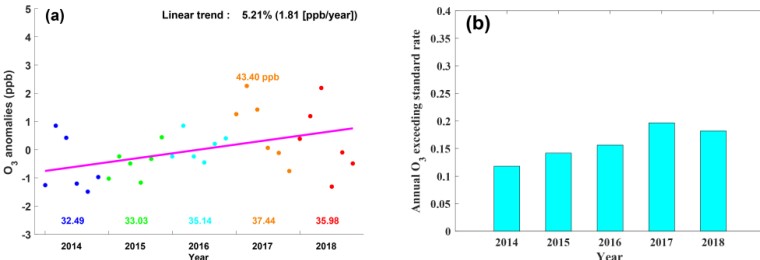


**Fig. 1. (a) Anomalies of monthly average $O_3$ concentration from April to September during**
**2014–2018. The purple solid line represents the linear fitted curve, and the color number**
**represents the annual (April–September) mean of $O_3$ concentration. (b) Annual**
**(April–September) variation in the days with $O_3$ exceeding the national standard.**

**3.1.2. Characteristics of $O_3$ variability based on the EOF analysis**



In order to further discuss the spatio-temporal distribution characteristics of the observed $O_3$
concentration, the EOF approach is used to uncover the relationship between the spatial
distribution and temporal variation. By removing the missing data for 17 days, $O_3$ concentrations
in 898 days are processed. The percentages of variance contribution for the first three patterns are
65.70 %, 13.80 % and 9.10 %, respectively. The significance tests of the EOF eigenvalue confirm
that the first three patterns are significantly separated. Approximately 88.60 % of the variability in
the original data is contained in these three patterns. In the first EOF pattern (EOF1), the observed
$O_3$ over the YRD changes similarly and the center of the variation is located in the middle of the
YRD (Fig. 2a). As shown in Fig. 2b, the time series of EOF1 presents a decreasing trend and
shows a high negative correlation with the time series of $O_3$ (R = −0.93). Therefore, to some extent,
the EOF1 time series variation can represent the daily mean $O_3$ variation during these periods.
Considering the negative values in EOF1, the EOF1 time series implies an increasing trend of
regional mean $O_3$ concentration. In addition, the relationships between the time series of EOF1
and different weather systems, as well as the meteorological factors have been investigated.
Weather systems include the WPSH and the East Asian summer monsoon, which are dominant
weather systems affecting the YRD. Both of them show a poor correlation with the EOF1 time
series ($R_{WPSH}$ = 0.13 and $R_{EASM}$ = 0.04). It indicates that the daily $O_3$ variation is too complex to
be comprehensively explained through the change in a single weather system. Furthermore, the
RH presents a good correlation with the EOF1 time series (R = 0.59). Han et al. (2020) also found
that RH is the most important factor affecting $O_3$ in the YRD. However, it is still unclear how the
change in different weather systems causes the variation in RH, and how the RH variation impacts
the other meteorological factors and $O_3$ accumulation.
In the second EOF pattern (EOF2), there is obvious east-west contrast. In contrast, the third
EOF (EOF3) pattern presents a notable south-north contrast. At the same time, the increasing trend
of EOF2 time series and the decreasing trend of EOF3 time series indicate that $O_3$ concentrations
in the west and northwest have risen from 2014 to 2018. It implies that a higher rate of $O_3$
increasing would occur in the northwest. As known, the variance contribution of EOF1 is 65.70 %
that is greater than EOF2 (13.80 %) and EOF3 (9.10 %). Therefore, the $O_3$ increasing in the whole
YRD region is the main trend.

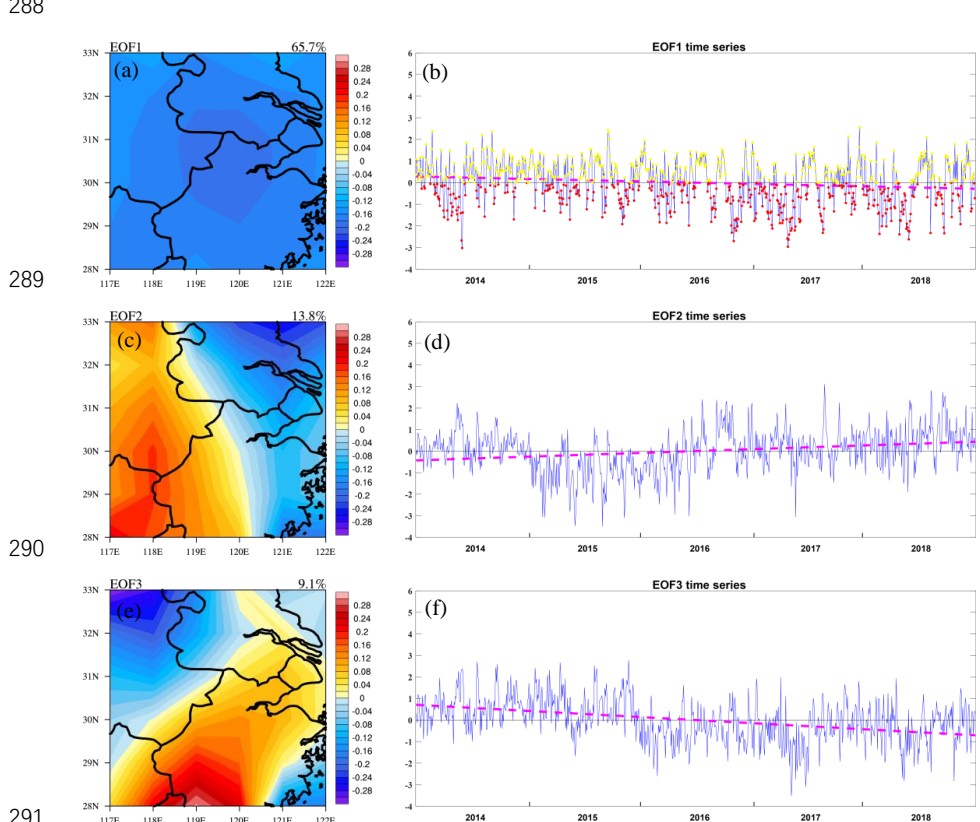




**Fig. 2. Three EOF patterns of O₃ concentration in the warm seasons from 2014 to 2018,**

**including the spatial pattern (a, c and e) and time coefficient (b, d and f). The percentage in**

**panels (a, c and e) is the variance contribution of each EOF mode. The orange dash line in**

**panels (b, d and f) represents the linear fitted curve.**

**3.2. Effects of meteorological conditions on O₃ concentration over the YRD region**

**3.2.1. Quantifying the effects of meteorological conditions**

With the primary pollutant emission being cut down, the O₃ increase might be affected by a

variety of factors, one of which was suggested to be the slowing down sink of hydroperoxy

radicals, related to the variation in PM$_{2.5}$ (Li et al. 2019). Yet, it is uncertain how meteorological

conditions influence this increasing trend. Yang et al. (2019) quantified the meteorological impact

on O₃ variation over the Pearl River Delta region using the meteorological adjustment. Similarly



to the methodology in Yang et al. (2019), we investigate the ozone increase over the YRD in the
warm seasons during 2014–2018. Fig. 3a shows the ambient $O_3$ variation from 2014 to 2018: i.e.
$O_3$ concentration increases form 2014, reaches the maximum in 2017, and maintains at a relatively
high level in 2018. After the meteorological adjustment, the increasing magnitude is lower than
the original one, implying that if the meteorological conditions remained unchanged over the 5
years, the increasing magnitude of ambient $O_3$ concentration would be lower. The meteorological
impact can be examined from the difference between the black solid and dashed lines in Fig. 3a.
We focus on periods from the middle of 2014 to the middle of 2018 when the difference is
negatively from the middle of 2014 to the middle of 2016 and positively large from middle of
2016 to the meddle of 2018. In 2017, the meteorological conditions increase the $O_3$ concentration
by about 1.20 ppb. However, in 2015, the meteorological conditions become unfavorable to the $O_3$
accumulation, leading to an $O_3$ reduction of 1.10 ppb. The meteorological conditions changed the
$O_3$ concentration by 2.81 ppb between the most favorable year (2017) and the most unfavorable
year (2015), which roughly corresponds to 9.62% ($\frac{max(MEO\ imapct)-min(MEO\ impact)}{O3(5\ year\ average)}$) of the
annual $O_3$ concentration.
In addition, we select the most influential meteorological factors to discuss their impact on
$O_3$ variation, including T, RH, sunshine duration and wind speed. As shown in Fig. 3b, RH is the
most crucial factor and its variation is similar to the variation in the total meteorological impact.
Han et al. (2020) also found that RH is the most influential factor in the central and south parts of
eastern China. The East Asian summer monsoon plays a key role in affecting the local RH, and
meanwhile it might bring a certain amount of $O_3$ from the south area. However, $O_3$ concentration
is highly negatively related to RH, which implies that the local chemical reaction might contribute
more to the $O_3$ accumulation than the regional transport. The contributions of the other three
factors are relatively insignificant.



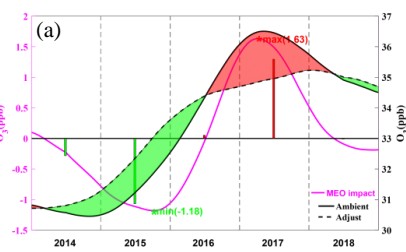 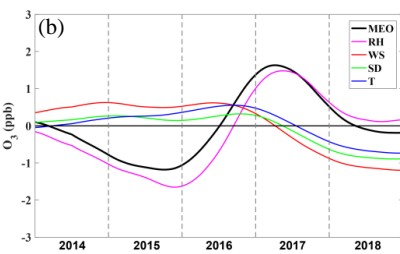

**Fig. 3. (a) 5-year trends of ambient O$_3$ (solid black line), meteorological adjusted O$_3$ (dashed black line), and the meteorological impact (pink line) over the YRD during 2014–2018. Periods with positive and negative meteorological impacts are shaded with red and green, respectively; red and green bars represent the O$_3$ increasing and decreasing caused by meteorological conditions in each year. (b) 5-year variations in the meteorological impact of different meteorological factors (MER), including relative humidity (RH), sunshine duration (SR), air temperature (T2) and wind speed (WP).**

**3.3. Dynamic processes of O$_3$ variation driven by synoptic circulations**

As discussed in section 3.2, the local meteorological factors have a great impact on the O$_3$ variation. However, to some extent, the variation in local meteorological factors is largely affected by the synoptic-scale weather circulations (Leibensperger et al. 2008; Fiore et al. 2003; Wang et al. 2016). For example, in summer the YRD is under a hot-wet environment controlled by the WPSH. While in winter it is under a cold-dry environment affected by the northwesterly flow caused by the Siberian High. The different weather systems under their corresponding SWPs have their unique meteorological characteristics. Moreover, even under one SWP, the location and intensity changes in a specific weather system can cause the changes in meteorological factors correspondingly (Gao et al. 2020).

**3.3.1. The main synoptic weather patterns in the warm season over the YRD**

Applying the PTT classification method, nine SWPs are identified for the warm seasons in the YRD. Due to the relatively large variance, the first dominant five types are selected, and the other four types are grouped as 'other types'. As shown in Table 1, SWP1, SWP2 and SWP4 are





dominant, accounting for 40.66%, 22.84% and 13.99% occurances, respectively. In contrast,
SWP3, SWP5 and other types are relatively lower, and their occurrence frequencies are 7.65%,
6.99% and 6.01%, respectively. Specifically, SWP1 is affected by the southeasterly flow
introduced by the WPSH. SWP2 is influenced by the northwesterly flow introduced by a
persistent high pressure. SWP4 is influenced by the southeasterly flow introduced by the WPSH
and a cyclone. SWP3 and SWP5 are affected by a cyclone and an anticyclone. For SWP1 and
SWP4, it is with high temperature and humidity affected by the southerly flow. But for SWP5,
because of the weak northerly flow which brings insufficient water vapor, the YRD is with high
temperature and low RH. SWP2 is with relatively lower temperature. SWP3 is under the control
of a cyclone and the strong upward motion, it is with weak SR and lower T.

**TABLE 1. The occurrence days and frequency, typical characteristics, regional mean $\pm$ the**
**standard error for temperature (T), relative humidity (RH), wind speed (WS) and solar**
**radiation (SR) and positive and negative days under each SWP. The > 0 and > 0.5 represent**
**the value of EOF1 time series more than 0 and 0.5, respectively. The < 0 and < 0.5 is on the**
**contrary.**

| Type and number of days (frequency ) | Typical characteristic of SWPs | Meteorological factors | Pos (>0 and >0.5) Neg (<0 and <0.5) (number of days) |
|---|---|---|---|
| SWP1 372 (41.43%) | Southwesterly flow introduced by WPSH | T (℃): 28.38 $\pm$ 4.94 <br> RH (%): 77.98 $\pm$ 10.44 <br> WS (m/s): 7.30 $\pm$ 0.54 <br> SR (W/m²): 1606.20 $\pm$ 537.77 | 194, 125 175, 112 |
| SWP2 209 (23.27%) | Northwesterly flow introduced by a continuant high pressure | T (℃): 26.40 $\pm$ 5.37 <br> RH (%): 73.97 $\pm$ 12.85 <br> WS (m/s): 7.28 $\pm$ 0.51 <br> SR (W/m²): 1615.00 $\pm$ 563.20 | 97, 57 110, 73 |
| SWP3 70 (7.80%) | an extratropical cyclone | T (℃): 25.41 $\pm$ 4.37 <br> RH (%): 86.80 $\pm$ 6.25 | 58, 45 12, 6 |

| | | WS (m/s): 7.33 ± 0.58 | |
| | | SR (W/m²): 959.73 ± 478.14 | |
| SWP4 128 (14.25%) | Southeasterly flow brought by WPSH and a southern cyclone system | T (℃): 29.29 ± 4.24 | |
| | | RH (%): 78.67 ± 8.51 | 82, 58 |
| | | WS (m/s): 7.11 ± 0.56 | 46, 30 |
| | | SR (W/m²): 1505.97 ± 538.96 | |
| SWP5 64 (7.13%) | The north China anticyclone system | T (℃): 28.08 ± 4.99 | |
| | | RH (%): 73.97 ± 12.03 | 23, 14 |
| | | WS (m/s): 7.22 ± 0.45 | 40, 24 |
| | | SR (W/m²): 1586.78 ± 479.65 | |
| others 55 (6.12%) | / | / | / |


### 3.3.2. Impacts of SWP change on $O_3$ concentration variation


We explore the impacts of SWP change on $O_3$ variation through combining the EOF1 mode.
As illustrated in section 3.1.2, the EOF1 mode is the dominant mode, and it implies the increase of
$O_3$ in the whole area is the main trend. Regarding EOF1 time series, it has a high correlation
coefficient with regional $O_3$ concentration (R = −0.93). In this study, we mainly focus on why $O_3$
concentration increases in the entire YRD region, rather than why the increases in $O_3$ differ
spatially inside the YRD. Therefore, we use the EOF1 time series as a proxy to present the
regional $O_3$ concentration. In Table 1, the positive phase (Pos) represents that the EOF1 time series
is more than 0 and it is not beneficial to the production and accumulation of $O_3$. On the contrary,
the negative phase (Neg) means the higher $O_3$ concentration. We extract the information by
comparing Neg with Pos to find the changes of each pattern. Yin et al. (2019) explored dominant
patterns of summer $O_3$ pollution and associated atmospheric circulation changes in eastern China.
Different from their study, we have analyzed the daily variation in SWPs, and can obtain the
change in atmospheric circulations more precisely.
In the five main SWPs, the EOF1 time series show a decrease trend during their occurrence
days in the warm seasons. It means the five main patterns tend to cause high ambient $O_3$



concentration through the change in SWPs. In addition, the SWP change includes both frequency
and intensity changes. We find that the frequency change in SWPs has less impact on the
inter-annual variation in $O_3$ levels than the intensity change in SWPs, which is consistent with the
results of Hegarty et al. (2007) and Liu et al. (2019). The contribution of intensity change and
frequency change will be further discussed in section 3.4. In the following, we will concretely
discuss the variation characteristics of SWPs and their impacts to the increase of $O_3$ in the YRD.

Fig. 4 shows the atmospheric circulations at 850 hPa and 500 hPa, and the box plot of

normalizing factors includes SR, T, TCC, RH, meridional wind at 850hPa (V850) and W (vertical
velocity) for SWP1_Pos and SWP1_Neg. As shown in Figs. 4a and 4b, the YRD is located at the
northwest of the WPSH, mainly affected by the southwesterly winds. Compared with the
SWP1_Pos, the range of WPSH is wider in the northwest area under SWP1_Neg, leading to the
strengthened southerly wind in the northwest, which results in higher temperature in this area. Due
to the weakening of V850, the water vapor transport acts in response from the south. RH shows a
decrease trend. At 500 hPa, a shallow trough is replaced by a slowly moving weak ridge, and the
downward motion would strengthen and last longer. The sink motion is favorable for the $O_3$
accumulation and $O_3$ photochemical reaction at the near surface. Besides, the decreasing water
vapor under the downward motion condition make the cloud cover hard to form. So, stronger solar
radiation hits the ground due to the less shelter from the cloud, further leading to higher air
temperature and stronger $O_3$ photochemical reaction.

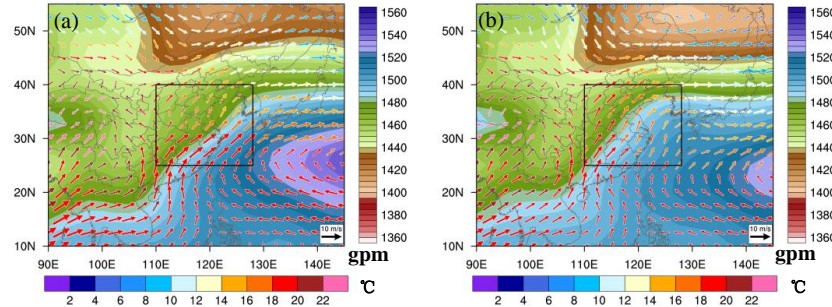


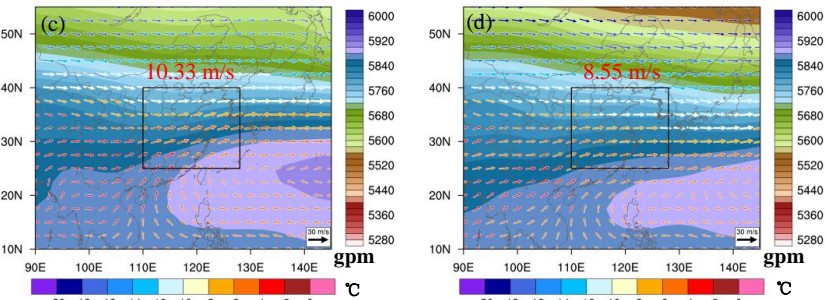

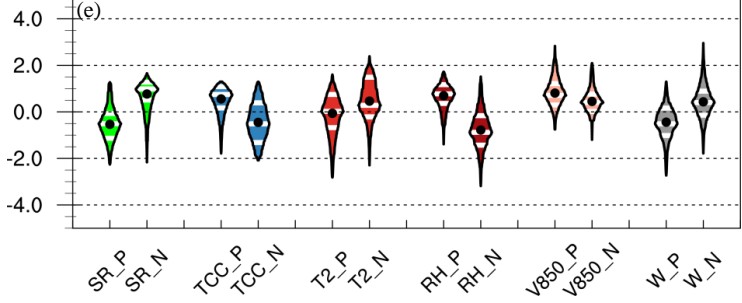

**Fig. 4. The geopotential height (shaded) and 850-hPa wind with temperature (color vector) under (a) SWP1_Pos and (b) SWP1_Neg. The geopotential height (shaded) and 500-hPa wind with temperature (color vector) under (c) SWP1_Pos and (d) SWP1_Neg. The red values represent regional average wind speed at 500 hPa in the zone around black lines. (e) The regional average meteorological factors under SWP1_Pos and SWP1_Neg, including SR, TCC, 2-m air temperature, RH, meridional wind at 850 hPa (V850) and W (vertical velocity). The boxed area in Figs.4a-4d encloses the YRD.**

Fig. 5 shows the atmospheric circulation structures at 850 hPa and 500 hPa, and the box plot of normalizing factors includes SR, T, TCC, RH, V850 and W for SWP2_Pos and SWP2_Neg. As shown in Figs. 5a and 5b, the YRD is affected by a continental high and the Aleutian low, characterized by northwesterly flow and a bit southwesterly flow. Compared with the SWP2_Pos, the northwesterly flow introduced by the continental high in SWP2_Neg is weaker. At the same time, as the Aleutian low moves southward slightly, the southwesterly flow can hardly bring water





vapor to the YRD, which leads to RH decreases in this area. The correlation between the EOF1
time series and 2-m air temperature under SWP2 ($R_{P2} = -0.41$) is closer than the correlation in the
whole period ($R_{all} = -0.24$). This implies that the weakening of the continent high plays an
important role in enhancing $O_3$ there. At 500 hPa, a trough is strengthened, leading to the stronger
downward motion. Just like SWP1, stronger downward motion and lower RH cause strong SR and
high air temperature. All these changes are beneficial to the $O_3$ formation and accumulation.

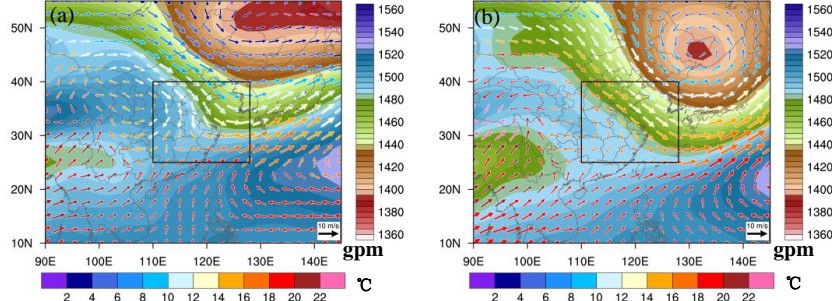


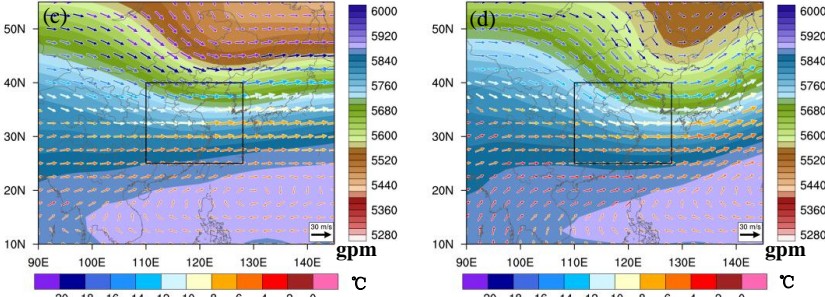







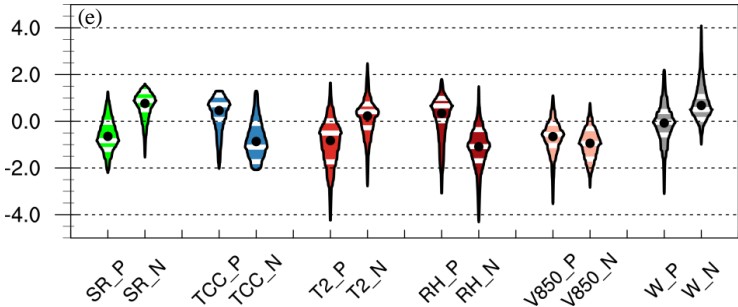

.

**Fig. 5. The geopotential height (shaded) and 850-hPa wind with temperature (color vector) under (a) SWP2_Pos and (b) SWP2_Neg. The geopotential height (shaded) and 500-hPa wind with temperature (color vector) under (c) SWP2_Pos and (d) SWP2_Neg. The red values represent regional average wind speed at 500 hPa in the zone around black lines. (e) The regional average meteorological factors under SWP2_Pos and SWP2_Neg, including SR, TCC, 2-m air temperature, RH, meridional wind at 850 hPa (V850) and W. The boxed area in Figs.5a-5d encloses the YRD.**

.

Fig. 6 shows the atmospheric circulations at 850 hPa and 500 hPa, and the box plot of normalizing factors includes SR, T, CC, RH, V850 and W for SWP3_Pos and SWP3_Neg. As shown in Figs. 6a and 6b, the YRD is controlled by an extratropical cyclone. Compared with the SWP3_Pos, the low pressure is lower and its location is slightly further eastward SWP3_Neg. Under this circumstance, the southerly flow at the bottom of the low pressure could hardly bring the water vapor to the YRD. At 500 hPa, the downward motion would be strengthened due to the strengthened trough. The intense downward motion and low RH result in less CC and strong SR, as well as high T, which are instrumental in high $O_3$ concentration.



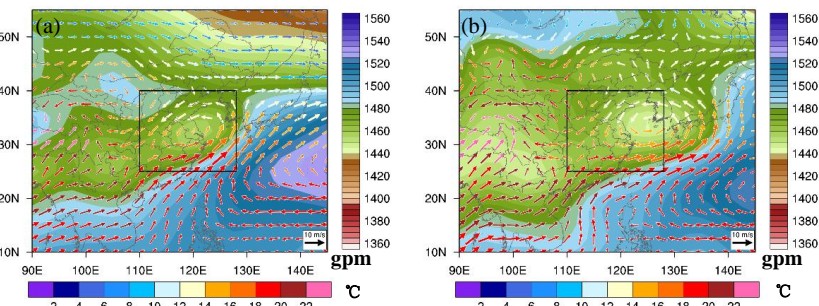


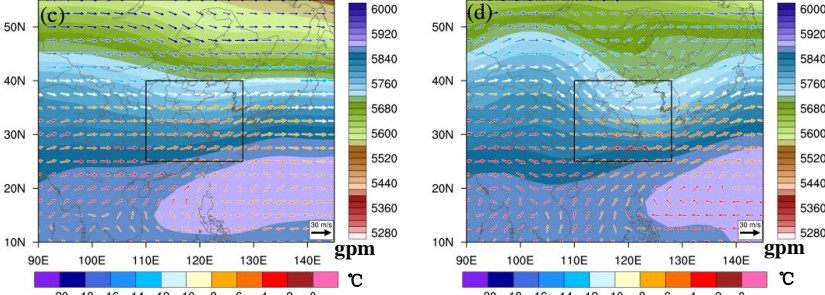



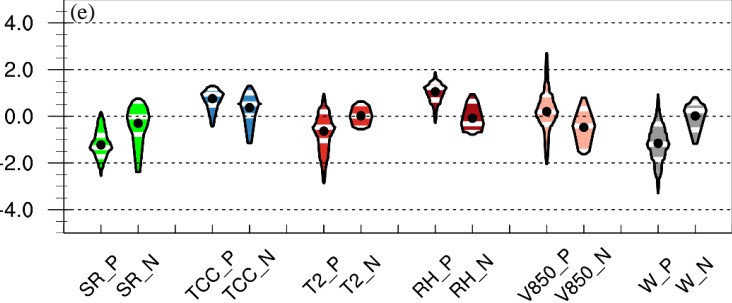


**Fig. 6. The geopotential height (shaded) and 850-hPa wind with temperature (color vector)**

**under (a) SWP3_Pos and (b) SWP3_Neg. The geopotential height (shaded) and 500-hPa**

**wind with temperature (color vector) under (c) SWP3_Pos and (d) SWP3_Neg. The red**

**values represent regional average wind speed at 500 hPa in the zone around black lines. (e)**

**The regional average meteorological factors under SWP3_Pos and SWP3_Neg, including SR,**





**TCC, 2-m air temperature, RH, meridional wind at 850 hPa (V850) and W. The boxed area**
**in Figs6a-6d encloses the YRD.**

Fig. 7 shows the atmospheric circulations at 850 hPa and 500 hPa, and the box plot of
normalizing factors includes SR, T, TCC, RH, V850 and W for SWP4_Pos and SWP4_Neg. As
shown in Figs. 7a and 7b, the southeasterly wins prevails in the YRD, which is caused by a
southern low pressure and the WPSH. Compared with the SWP4_Pos, the southern low pressure
and southeasterly flow is weaker in SWP4_Neg, and thus it brings less water vapor to the YRD. At
500hPa, a shallow trough strengthens, causing the strong sink motion. High temperature, strong
SR and low RH caused by the low V850 and downward motion are favorable for the $O_3$
accumulation.

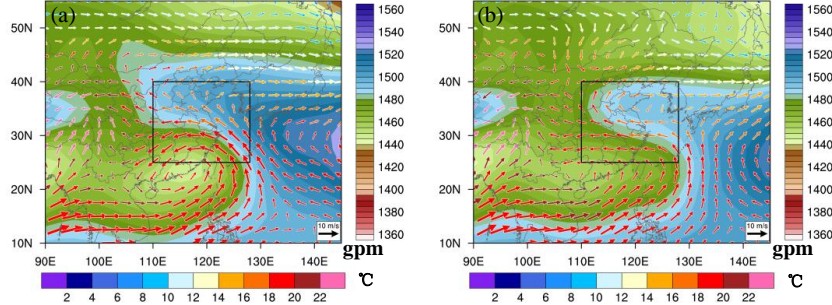


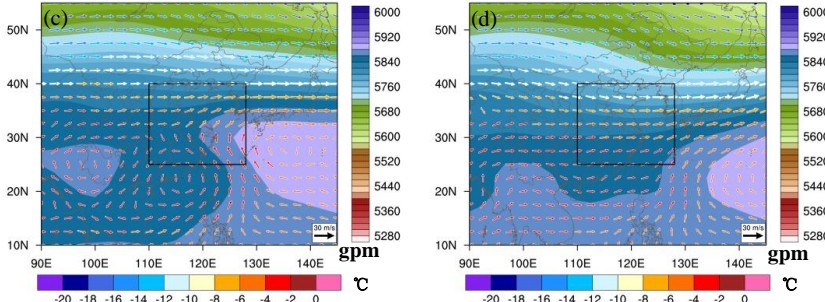





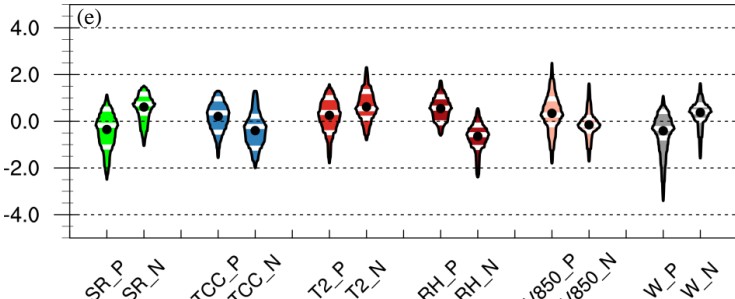


**Fig. 7. The geopotential height (shaded) and 850-hPa wind with temperature (color vector)**
**under (a) SWP4_Pos and (b) SWP4_Neg. The geopotential height (shaded) and 500-hPa**
**wind with temperature (color vector) under (c) SWP4_Pos and (d) SWP4_Neg. The red**
**values represent regional average wind speed at 500 hPa in the zone around black lines. (e)**
**The regional average meteorological factors under SWP4_Pos and SWP4_Neg, including SR,**
**TCC, 2-m air temperature, RH, meridional wind at 850 hPa (V850) and W. The boxed area**
**in Figs.7a-7d encloses the YRD.**

Fig. 8 shows the atmospheric circulations at 850 hPa and 500 hPa, and the box plot of

normalizing factors includes SR, T, TCC, RH, V850 and W for SWP5_Pos and SWP5_Neg. As
shown in Figs. 8a and 8b, the YRD is controlled by the north China anticyclone, characterized by
the northeasterly and the southwesterly winds. Compared with the SWP5_Pos, the high pressure
in the SWP5_Neg is weaker and the northeasterly flow would act in response. The weakened cold
sea flow makes the air warmer and dryer. At 500hPa, a trough controlling the YRD would be
strengthened. The downward motion would become strong correspondingly. The favorable for the
$O_3$ accumulation.




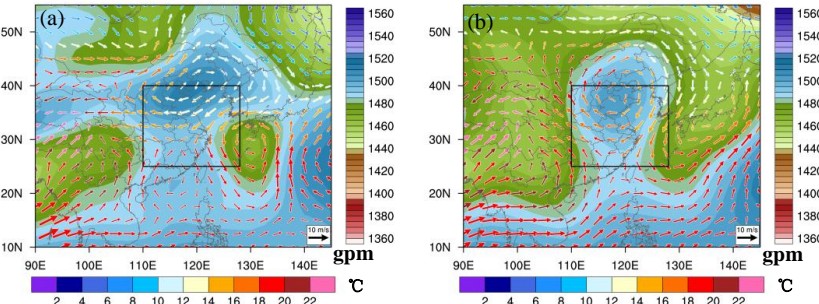



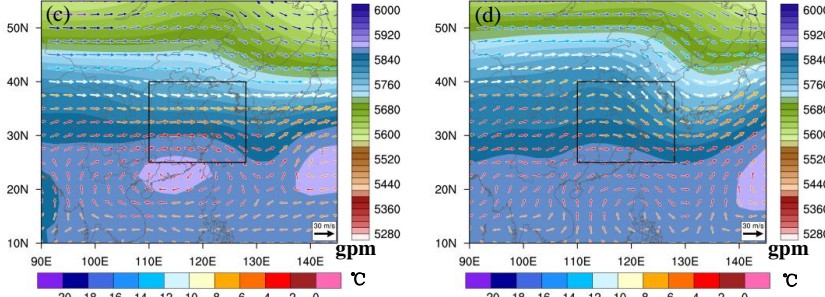

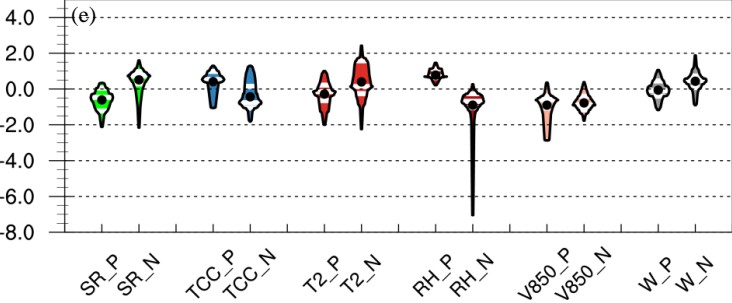


**Fig. 8. The geopotential height (shaded) and 850-hPa wind with temperature (color vector)**
**under (a) SWP5_Pos and (b) SWP5_Neg. The geopotential height (shaded) and 500 hPa**
**wind with temperature (color vector) under (c) SWP5_Pos and (d) SWP5_Neg. The red**
**values represent regional average wind speed at 500 hPa in the zone around black lines. (e)**
**The regional average meteorological factors under SWP5_Pos and SWP5_Neg, including SR,**





**TCC, 2-m air temperature, RH, meridional wind at 850 hPa (V850) and W. The boxed area**

**in Figs.8a-8d encloses the YRD.**

**3.4. Indicators for reconstructing inter-annual $O_3$ variation affected by synoptic-scale atmospheric circulation**

Due to the similar variations in regional mean $O_3$ concentration and EOF1 time series, we have reconstructed the inter-annual EOF1 time series to replace the regional mean $O_3$ concentration by taking into account either frequency-variation-only or both frequency and intensity variations in SWPs, which are EOF1 time series (Fre) and EOF1 time series (Fre + Int), respectively. The observed and reconstructed inter-annual EOF1 time series in 2014–2018 in the whole region are shown in Fig. 9. Obviously, the frequency changes in SWPs almost have no impact on the $O_3$ variability in the entire YRD. Regarding the intensity change, the fitting curve would be closer to the EOF1 time series. Hegarty et al. (2007) and Liu et al. (2019) reconstructed the inter-annual $O_3$ level in the northeastern United States and the northern China using the same method as ours. Moreover, they defined the intensity change in SWPs using the domain-averaged sea level pressure and the pressure of the lowest-pressure system. As illustrated by Hegarty et al. (2007), the correlation under Pattern V is poor. It indicates we should select the SWPIIs under each pattern according to their unique characteristics on high $O_3$ concentration. We select six SWPIIs: the maximum pressure in zone 1 (25 °N–40 °N, 110 °E–130 °E) and zone 2 (20 °N–50 °N, 90 °E–140 °E), the minimum pressure in zone 1 (25 °N–40 °N, 110 °E–130 °E) and zone 2 (20 °N–50 °N, 90 °E–140 °E), and the average pressure in zone 1 (25 °N–40 °N, 110 °E–130 °E) and zone 3 (10 °N –40 °N, 110 °E–130 °E). As shown in Table 2, the SWPII for the maximum pressure in zone 1 has a relative high correlation between SWP3 and SWP5, and the SWPII for the maximum pressure in zone 2 has a relative high correlation between SWP1 and SWP4. The annual EOF1 time series anomalies show a relative good correlation with the maximum pressure. It is because the maximum pressure reflects the wind speed affecting the water vapor transport under this pattern. Compared with SWP3 and SWP5, the synoptic system is larger than the classification region for SWP1 and SWP4. So it can represent the SWPII more precisely in a large region. Under SWP2, when $O_3$ concentration tends to be at a high level, a cold continent high behind the





YRD would tend to weaken. Therefore, we select the average height in zone 3 to represent the
SWPII under SWP2. From Table 2, we can know it has better reconstructed curve when we
selected different SWPIIs according to the characteristics of high $O_3$ level under each pattern.
Above all, the intensity change in SWP is more important to the inter-annual $O_3$ variation than the
frequency change.

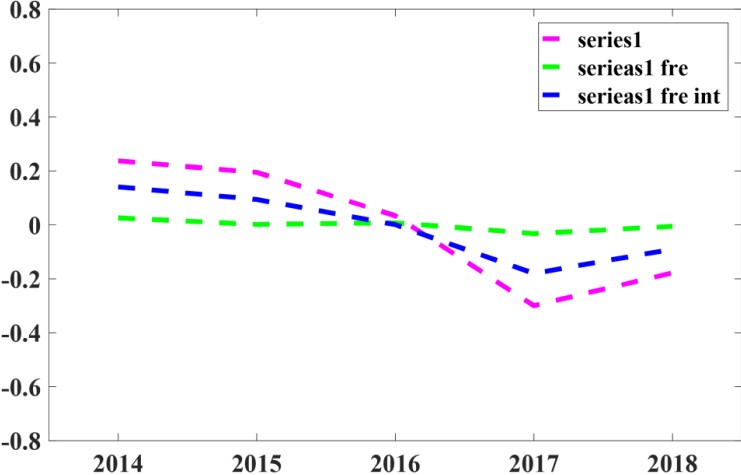


**Fig. 9. The original and reconstructed inter-annual EOF1 time series trend based on SWP**
**frequency and intensity variations. The pink curve represents the original inter-annual**
**EOF1 time series, whereas the green and blue lines are the trends of reconstructed**
**inter-annual EOF1 time series according to the frequency-variation-only and both frequency**
**and intensity variations in SWPs, respectively.**

**TABLE 2. Correlation coefficients between EOF1 time series and different SWPIIs under**
**each SWP.**

| Type | $Z_{1\text{-ave}}$ | $Z_{1\text{-max}}$ | $Z_{1\text{-min}}$ | $Z_{2\text{-min}}$ | $Z_{2\text{-max}}$ | $Z_{3\text{-ave}}$ |
|------|------|------|------|------|------|------|
| SWP1 | -0.47 | -0.29 | -0.35 | -0.33 | -0.60 | -0.32 |
| SWP2 | -0.14 | -0.08 | 0.02 | -0.07 | -0.09 | -0.40 |
| SWP3 | 0.28 | 0.61 | 0.03 | 0.005 | 0.43 | -0.60 |
| SWP4 | -0.14 | -0.03 | -0.17 | -0.22 | 0.78 | -0.38 |





| SWP5 | 0.52 | 0.76 | 0.39 | 0.56 | 0.72 | 0.58 |


**4. Conclusions and discussions**
In this study, we discussed the meteorological influences on the $O_3$ variation in the warm
seasons during 2014–2018 in the YRD, China. Specifically, we analyzed the $O_3$ spatio-temporal
distribution characteristics, quantified the contribution of meteorological conditions to $O_3$
variations, explored how the change in SWPs and corresponding meteorological factors lead to $O_3$
increase over 2014-2018, and quantitatively analyzed the impact of SWP frequency and intensity
on the inter-annual $O_3$ variation. The main conclusions are as follows.
The annual regional averaged $O_3$ concentrations from 2014 to 2018 in the YRD are 32.49,
33.03, 35.14, 37.44 and 35.98 ppb, respectively, with a significantly increasing rate of 1.81 ppb
year$^{-1}$ (5.21% year$^{-1}$). At the same time, the total number of days on which $O_3$ concentration
exceeds the national standard also increases with year in a similar pattern. Through the EOF
analysis of $O_3$ in space and time, three dominant modes were identified. The first mode is the most
dominant mode, accounting for 65.7% of $O_3$ variation, implying that the $O_3$ increasing in the entire
YRD is the main tendency. A high correlation coefficient between the EOF1 time series and RH
($R_{rh} = 0.59$) indicates that RH is the most influential factor leading to the $O_3$ increase.
We quantified the influence of meteorology on inter-annual variation and trend of $O_3$ over the
YRD from 2014-2018, and found that the influence could lead to a regional $O_3$ increase by 2.81
ppb at most. Especially, RH plays the most important role in modulating the inter-annual $O_3$
variation. Moreover, in order to explore connections between the $O_3$ variation and synoptic
circulations, nine types of SWPs were objectively identified based on the PTT method, and five
main types were selected to correlate with the EOF1 time series. We found that the variation in all
SWPs over 2014-2018 are favorable to $O_3$ increase in that period. The variation in SWP intensity
include the WPSH weakening and northward extending under SWP1, a continent high weakening
under SWP2, an extratropical cyclone strengthening under SWP3, the southern low pressure
weakening and the WPSH weakening under SWP4, and the north China anticyclone weakening
under SWP5. All these changes prevent the water vapor from being transported to the YRD and
increase air temperature in YRD. In addition, the downward motions strengthen in the YRD,





which is behind the trough and in front of the ridge due to the strengthening of the ridge and
trough, leading to less cloud cover and stronger SR. All of these are favorable to $O_3$ formation and
accumulation.

We found that the change in SWPs intensity is more important to the $O_3$ increase over

2014-2018 than that in SWPs frequency. We further reconstructed the EOF1 time series by
considering different SWPIIs due to the unique characteristics of each SWP. The results are better
than those in Hegarty et al. (2007) and Liu et al. (2019) who used the same SWPIIs in all SWPs.

In summary, this study quantified the inter-annual variation and increasing rate of $O_3$ in the

YRD, China, and explored the connection between SWP variations and the $O_3$ increase. It
provides an enhanced understanding of response of $O_3$ variation to changes in SWPs from year to
year and thus this understanding may be insightful to planning strategies for $O_3$ pollution control.

**Authorship contribution statement**

**Da Gao:** Conceptualization, Data curation, Formal analysis, Meteorology, Investigation,

software, Writing – original draft, Writing – revision. **Min Xie**: Conceptualization, Methodology,
Writing – revision, Project administration, Funding acquisition. **Jane Liu:** Formal analysis,
Meteorology, Writing – revision. **Tijian Wang**: Formal analysis, Funding acquisition. **Chaoqun**
**Ma**: Formal analysis, Meteorology. **Haokun Bai**: Formal analysis, Meteorology. **Xing Chen**:
Formal analysis.

**Declaration of competing interest**

The authors declare that they have no known competing financial interests or personal

relationships that could have appeared to influence the work reported in this paper.

**Acknowledgements**
This work was supported by the National Key Research and Development Program of China
(2018YFC0213502, 2018YFC1506404).

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
