# Peer review of "Ozone Variability Induced by Synoptic Weather Patterns"

_Atmospheric Chemistry and Physics, 2020_

## Referee Comment (RC1) · Anonymous Referee #2 · 1 Dec 2020

This paper discussed the meteorological influence on the increase in ozone concentration in the YRD china. Abundant analysis methods were used to try to figure out the reason to the increase in the ozone concentration. The results are some helpful to ozone pollution control and prediction. I have some comments in the following to improve this paper.

General comments: You attributed the effect of low RH on the increase in the ozone concentration to the strong solar radiation and high temperature in all kinds of SWPs. However, why does RH show a significant correlation with ozone concentration instead of the more direct meteorological factors temperature and radiation? Moreover, as you

said, the lack of clouds contribute much to the high concentration of ozone. Why not analyze the impact of cloud property (such as cloud fraction, cloud thickness, cloud height, cloud liquid content) in the meteorological dataset? Cloud is the direct impact factor probably. Through the results and discussion section, they are almost qualitative description. Additional quantitative analysis and discussion are needed to make your conclusion more significant and scientific. The discussions in S3.3.2 about the impacts of SWP on ozone concentration are too similar for five SWPs. They all results in the downward motion, high temperature, strong radiation. I suggest to pay more attention to the difference in the impact among SWPs.

Line 115: How many sites in total in 26 cities were used in your research? Or you used the mean concentration for each city? Line 125: How many missing data in your dataset? Can you evaluate the influence of these missing data on your conclusion? Line 130: Please list the number of coefficients you used in this function. Line 244: I suggest to use all data instead of monthly mean over 26 cities to do linear fitting because some extreme high concentration in several cities may change the fitting results. Please show the fitting function and correlation coefficient. Line 272: How did you define the coefficient of meteorological factors like WPSH, EASM? How did you calculate the correlation between meteorological factors and ozone concentration? Fig3: The abbreviations are different in the figure and captions. Line 352: Here you said "SWP1 is affected by the southeasterly flow...", while "Southwesterly flow" for SWP1 in the Table 1. Section 3.3.1: It is better to show these six SWPs in figure addition to the Table 1, at least in the supplementary. Table 1: What do the meteorological factors mean? Regional average during all warm seasons? Line 379: How did you analyze the daily variation? What is the influence on the result? Line 384: How can you get the conclusion that frequency change has less impact than the intensity change? Please add quantitative evaluation. Line 393: Please describe the difference in WPSH using some representative index like WPSH index, ridge position, instead of the puzzled word "wider". Line 399: If the downward airmass comes from ocean with abundant water vapor, although the cloud is hard to form, the RH on the surface possibly increases.

How can you explain the negative correlation between surface RH and ozone concentration? same question for the explanation of other SWPs. Line 511: I don't think it is obvious that frequency changes have on impact. It looks that the contribution from frequency changes is comparable to that from intense changes according to Fig9. Could you give more explanation or evidence? Line 517: What are patter V? Line 517: What is the definition of "SWPII"? How did you calculate it? It is better to show the number for each SWP. Line 551: I did not find much quantitatively analysis in your discussion, but it should be needed.

---

## Referee Comment (RC2) · Anonymous Referee #1 · 1 Dec 2020

General comments: In this manuscript, the authors focus on the inter-annual variations of warm seasons (April–September) ozone over YRD, China from 2014 to 2018. The relations between the inter-annual ozone and synoptic-scale circulations and the associated meteorological controlling factors were revealed. The authors highlight five dominant synoptic weather patterns (SWPs) in the warm seasons in YRD using the t-mode principal component analysis and reconstructed the inter-annual O3 variation based on SWPs frequency and intensities. The analysis is mostly sound, especially on inter-annual ozone variations impact by SWPs induced meteorological factors. But some analysis need deeper explanation in physical or photochemical principals, and some conclusions need more robust supports. My specific suggestions and comments

are as follow. Specific comments: 1.In the title and content of the paper, I feel the "Dynamic Processes" could not give a direct and effective cognition to reader. I suggest "Ozone Variability in Warm Seasons of 2014–2018 over the Yangtze River Delta, China induced by synoptic patterns" or similar titles should be better. 2. In abstract line 37-41 and also in the context, the 2 sentences may conflict. I am not sure "the strengthening of the ridge and trough in the westerlies" is conflict with "the weakening of the continental high under SWP2" and "the southern low pressure weakening and WPSH weakening under SWP4, and the north China anticyclone weakening under SWP5.". In comparison with the similar previous studies (Han, et al., 2020; and Gao et al., 2020), this paper is not clear in spatial distribution of pressure and lack of clear pictures in synoptics. 3.In figure 1, "43.40" need mention in the context. 4.In the EOF analysis, the spatial distributions of EOF1 are generally negative and time series of EOF1 presents a decreasing trend. Actually, the O3 generally increase all over YRD in recent years. So, I suggest multiply -1 with spatial distributions of EOF1 and time series of EOF1 make the statement easy to follow. 5.The authors reveal RH is the key factor dominating inter-annual variations of ozone, and indicate that its unclear in the relations between RH and ozone in previous study. I suggest the authors gives a further clear explanation of the RH effects on ozone. RH could related to the cloud cover (solar radiation), stable of air in BL and so on. In figure 3b, sunshine duration (may related to cloud cover?) is not important in ozone inter-annual variation, and opposite to RH, which may implicate that stable of air (accumulation of air pollutants) is important? 6.In line 402, "the cloud cover hard to form" should be "the cloud hard to form." 7.What is the unit in figure 4 of W (vertical velocity), m/s or Pa/s? 8.In line 427-428, the sentence "At 500 hPa . . .." should indicate the area of downward motion. 9.What is SR in fig. 4 etc.? SD? 10.From figure 4-8, a summary table with values of meteorological factors in 5 SWPs could be better than sub-figure (e) for comparisons. 11.In section 3.4, I wonder why do you reconstruct the EOF1 time series? It could be more valuable to reconstruct the inter-annual variations of ozone concentration based on SWPs frequencies and intensities. And What's SWPIIs? Technical corrections: 1.In the caption

of figure 2, are they "orange dash line"? Pink? 2.There are several typo need carefully check. For example, meddle in line 313; "wins" in line 468; "SR" in 336 could be SD.
* * *

---

## Author Comment (AC2) · 9 Feb 2021

General comments: In this manuscript, the authors focus on the inter-annual variations of warm seasons (April–September) ozone over YRD, China from 2014 to 2018. The relations between the inter-annual ozone and synoptic-scale circulations and the associated meteorological controlling factors were revealed. The authors highlight five dominant synoptic weather patterns (SWPs) in the warm seasons in YRD using the t-mode principal component analysis and reconstructed the inter-annual O3 variation based on SWPs frequency and intensities. The analysis is mostly sound, especially on inter-annual ozone variations impact by SWPs induced meteorological factors. But

some analysis need deeper explanation in physical or photochemical principals, and some conclusions need more robust supports. My specific suggestions and comments are as follow.

Thanks for your comments and suggestions. All your comments and suggestions are very important. They have directive significance to our paper and research work.

Specific comments: 1.In the title and content of the paper, I feel the "Dynamic Processes" could not give a direct and effective cognition to reader. I suggest "Ozone Variability in Warm Seasons of 2014–2018 over the Yangtze River Delta, China induced by synoptic patterns" or similar titles should be better.

Thanks for your comments. "Dynamic Processes" originally emphasized the meteorological influences on ozone variability. In order to avoid confused cognition of readers, the old title is replaced by "Ozone Variability Induced by Synoptic Weather Patterns in Warm Seasons of 2014-2018 over the Yangtze River Delta, China". Please see the new title in the new revised manuscript.

2. In abstract line 37-41 and also in the context, the 2 sentences may conflict. I am not sure "the strengthening of the ridge and trough in the westerlies" is conflict with "the weakening of the continental high under SWP2" and "the southern low pressure weakening and WPSH weakening under SWP4, and the north China anticyclone weakening under SWP5."

Thanks for your comments. The strengthening of ridge and trough in the westerlies are associated with the strengthening of dominated weather systems. However, under SWP2, 4 and 5, changes in troughs and ridges are not associated with changes in the continental high, the WPSH and the north China anticyclone. Specifically, the trough and ridge strengthening are associated with the Aleutian low shifting southward under SWP2, the southern low pressure weakening under SWP4 and Japan low pressure appearance under SWP5. To clarify the above findings, we add the following explanations on lines 463, 499 and 515 in the new revised manuscript.

Line 463: At 500 hPa, a trough located at approximate 120°E–125°E is strengthened associated with Aleutian low shifting southward,

Line 499: At 500 hPa, a shallow trough located at about 125°E strengthens associated with weakening of the southern cyclone pressure,

Lines 515: At 500 hPa, a trough located at about 130°E controlling the YRD strengthens associated the Japan low pressure appearance.

In comparison with the similar previous studies (Han, et al., 2020; and Gao et al., 2020), this paper is not clear in spatial distribution of pressure and lack of clear pictures in synoptics.

Thanks for your comments. In order to show clear pictures in synoptic, specific figures of atmospheric circulation at 850 hPa under each SWP are added in the supplement. As shown in Fig. 1, SWP1 is under control of the southwesterly flow introduced by the WPSH. SWP2 is influenced by the northwesterly flow introduced by a continental high pressure and the Aleutian low pressure. SWP4 is influenced by the southeasterly flow introduced by a cyclone and an anticyclone. The above findings are added on lines 376-379 in the new revised manuscript as well.

These figures are similar with figures in Pos phase or Neg phase under each SWP, and we primarily explore the changes in atmospheric circulation between Pos and Neg phase of the SWPs. Therefore, these figures are only added in the supplement.

3.In figure 1, "43.40" need mention in the context.

Thanks for your comments. "43.40 ppb" represents the highest monthly mean O3 concentration value during the warm seasons in 2014-2018. Fig. 2a primarily shows the increasing trend during this period, so it is inappropriate to illustrate this maximum number in the figure. In the new revised manuscript, "43.40" is deleted.

4.In the EOF analysis, the spatial distributions of EOF1 are generally negative and

СЗ

time series of EOF1 presents a decreasing trend. Actually, the O3 generally increase all over YRD in recent years. So, I suggest multiply -1 with spatial distributions of EOF1 and time series of EOF1 make the statement easy to follow.

Thanks for your suggestions. The spatial distributions and time series of EOF1 mode have been multiplied -1 in Fig. 3. In addition, we correspondingly change Fig. 4 and other illustrations. Please see Figs.2 and 9, and the words on lines 398-400 in the new revised manuscript. The revisions are listed as below:

Lines 398-400: the positive phase (Pos) represents that the EOF1 time series is more than 0 and it is beneficial to the production and accumulation of O3. On the contrary, the negative phase (Neg) corresponds low O3 concentration.

5. The authors reveal RH is the key factor dominating inter-annual variations of ozone, and indicate that its unclear in the relations between RH and ozone in previous study. I suggest the authors gives a further clear explanation of the RH effects on ozone. RH could related to the cloud cover (solar radiation), stable of air in BL and so on. In figure 3b, sunshine duration (may related to cloud cover?) is not important in ozone inter-annual variation, and opposite to RH, which may implicate that stable of air (accumulation of air pollutants) is important?

Thanks for your comments. We re-quantify the meteorological factors impact on the O3 variation, and stress influential mechanism on O3 variation from dominated meteorological factors.

In section 3.2.1, we quantify the meteorological impact on the O3 variation using meteorological adjustment method. In the original manuscript, we adopted sunshine duration, air temperature at 2m, wind speed at 10m and relative humidity as the input factors. According to the above suggestions, in order to clarify the effects of relative humidity (RH) on O3, we replace the sunshine duration with solar radiation (SR) and add the low cloud cover (LCC). There are two reasons for selecting LCC to analysis. Firstly, low clouds are more effective at blocking out sunlight (SR) than medium and high clouds. Secondly, LCC has the higher correlation coefficient with SR than total cloud cover, medium cloud cover and high cloud cover. As shown in Fig.5, RH is the most crucial factor and its variation is similar to the variation in the total meteorological impact. In addition, SR and LCC also play important roles and have large impacts on O3 variation. RH can impact O3 concentration in two ways. One is gas phase H2O reacting with O3 (O3 + H2O + hv = O2 + 2OH). The other is its influencing on clouds and thereby shielding SR. During this process, specifically, under low RH circumstance, the reactions between water vapor and O3 are inhibited. Moreover, low RH leads to less cloud cover, and thereby there is more intensive SR. Strong SR can enhance O3 chemical reaction.

In a word, RH, SR and LCC all have important effects on O3 variation. Among them, RH plays the most significant role in modulating the inter-annual O3 variation. Low RH prevents O3 to react with gas phase H2O. Moreover, low RH caused by vertical downward motions results in less LCC and intensive SR, which can enhance the O3 chemical reactions and lead to higher O3 concentrations. The above-mentioned specific discussions have been added in section 3.2.1 in the new revised manuscript.

6.In line 402, "the cloud cover hard to form" should be "the cloud hard to form."

Thanks for your suggestions. "the cloud cover hard to form" is changed to be "hinder cloud formation". Please see line 440 in the new revised manuscript

7. What is the unit in figure 4 of W (vertical velocity), m/s or Pa/s?

The unit of W (vertical velocity) is "Pa/s". In the new revised manuscript, the sub-figures (e) in Fig. 4, 5, 6, 7, 8 have been replaced by Table 3, and the unit of W "Pa/s" have been added in the Table 3. Please see line 527 in the new revised manuscript.

8.In line 427-428, the sentence "At 500 hPa . . .." should indicate the area of downward motion.

Thanks for your suggestions. As shown in Fig. 6a and b, the northwest YRD area in

red box is located behind the strengthening trough. According to the potential tendency equation, downward motions are usually located behind the trough. Thus, in this case, the northwest YRD area is associated with stronger downward motion. The above discussion is added in the new revised manuscript. Please see lines 462-464.

9.What is SR in fig. 4 etc.? SD?

Thanks for your suggestions. SR and SD represent the solar radiation and sunshine duration, respectively. Solar radiation reanalysis data and sunshine observation data are acquired from the ERA-interim dataset and the air quality real-time publishing platform. SR is usually regarded as the directly influential factor of O3 formation. Therefore, in the section 3.2.1, we quantify the effect of meteorological conditions by using SR. In the new revised manuscript, SD is not used any more

10. From figure 4-8, a summary table with values of meteorological factors in 5 SWPs could be better than sub-figure (e) for comparisons.

Thanks for your suggestions. We change sub-figures (e) in Fig. 4, 5, 6, 7, 8 into a summary table. Table 1 shows regional mean  $\pm$  the standard error of meteorological factors in Pos phase and Neg phase and their difference (Pos minus Neg) under each pattern. The meteorological factors include relative humidity (RH), solar radiation (SR), air temperature (T2), low cloud cover (LCC), total cloud liquid water (TCLW), zonal wind speed at 850 hPa (V850) and vertical motion (W). Table 1 is different to show in this text, So It present in the form of figures and is added after Fig. 6.

RH, SR and T2 are dominated meteorological factors affecting O3 variation. V850 is an important element of bringing water vapor to the YRD and result in RH variations. Moreover, under the condition of vertical upward or downward motion (W), RH would change low cloud cover (LCC) and total cloud liquid water (TCLW), leading to the variation of SR.

11.In section 3.4, I wonder why do you reconstruct the EOF1 time series? It could be

more valuable to reconstruct the inter-annual variations of ozone concentration based on SWPs frequencies and intensities. And What's SWPIIs?

Thanks for your suggestions. We reconstruct the EOF1 time series to replace the regional mean O3 concentration. There are two reasons for this decision. Firstly, the time series of EOF1 shows a high negative correlation with the O3 time series (R = -0.98). More importantly, we primarily focus on why O3 concentration increases in the entire YRD region, rather than why the increases in O3 differ spatially inside the YRD. Therefore, it is more appropriate to reconstruct EOF1 time series than O3 time series.

SWPIIs represent synoptic weather pattern intensity indexes. They are defined as maximum geopotential height in zone  $1(25^{\circ}N-40^{\circ}N, 110^{\circ}E-130^{\circ}E)$  for SWP3 and SWP5, maximum geopotential height in zone 2 ( $20^{\circ}N-50^{\circ}N, 90^{\circ}E-140^{\circ}E$ ) for SWP1 and SWP4, and average geopotential height in zone 3 ( $10^{\circ}N - 40^{\circ}N, 110^{\circ}E-130^{\circ}E$ ) for SWP2, according to their high correlation coefficients with EOF1 time series under each SWP. Especially, zone1, 2 and 3 were selected in term of location of dominated weather systems under each SWP. Please see lines 255-257 and 559-570 in the new revised manuscript.

Technical corrections: 1.In the caption of figure 2, are they "orange dash line"? Pink? 2.There are several typo need carefully check. For example, meddle in line 313; "wins" in line 468; "SR" in 336 could be SD. Thanks for your suggestions. The abovementioned typos have been corrected. For example, "orange", "meddle" and "wins" are modified as "pink", "middle", "winds". Please see lines 314, 332 and 495 in the new revised manuscript. In the new revised manuscript, Sunshine duration (SD) is not used any more, as mentioned in the response to the specific comment 9.

Fig. 1. The geopotential height (shaded) and 850 hPa wind with temperature (color vector) under (a) SWP1, (b) SWP2, (c) SWP3, (d) SWP4, (c) SWP5. The boxed area in Figs.1a-e encloses the

Fig. 2. (a) Anomalies of monthly average O3 concentration from April to September during 2014–2018. The purple solid line represents the linear fitted curve, and the color number represents the annual (April–September) mean of O3 concentration.

Fig. 2.

---

## Author Response (AR1)

**Response to the comments of two referees**

**Response to the comments of Referee #1:**

General comments: In this manuscript, the authors focus on the inter-annual variations of warm seasons (April–September) ozone over YRD, China from 2014 to 2018. The relations between the inter-annual ozone and synoptic-scale circulations and the associated meteorological controlling factors were revealed. The authors highlight five dominant synoptic weather patterns (SWPs) in the warm seasons in YRD using the t-mode principal component analysis and reconstructed the inter-annual O3 variation based on SWPs frequency and intensities. The analysis is mostly sound, especially on inter-annual ozone variations impact by SWPs induced meteorological factors. But some analysis need deeper explanation in physical or photochemical principals, and some conclusions need more robust supports. My specific suggestions and comments are as follow.

Thanks for your comments and suggestions. All your comments and suggestions are very important. They have directive significance to our paper and research work.

Specific comments:

1.In the title and content of the paper, I feel the "Dynamic Processes" could not give a direct and effective cognition to reader. I suggest "Ozone Variability in Warm Seasons of 2014–2018 over the Yangtze River Delta, China induced by synoptic patterns" or similar titles should be better.

Thanks for your comments. "Dynamic Processes" originally emphasized the meteorological influences on ozone variability. In order to avoid confused cognition of readers, the old title is replaced by "Ozone Variability Induced by Synoptic Weather Patterns in Warm Seasons of 2014-2018 over the Yangtze River Delta, China". Please see the new title in the new revised manuscript.

2. In abstract line 37-41 and also in the context, the 2 sentences may conflict. I am not sure "the strengthening of the ridge and trough in the westerlies" is conflict with "the weakening of the continental high under SWP2" and "the southern low pressure weakening and WPSH weakening under SWP4, and the north China anticyclone weakening under SWP5.".

Thanks for your comments. The strengthening of ridge and trough in the westerlies are associated with the strengthening of dominated weather systems. However, under SWP2, 4 and 5, changes in troughs and ridges are not associated with changes in the continental high, the WPSH and the north China anticyclone. Specifically, the trough and ridge strengthening are associated with the Aleutian low shifting southward under SWP2, the southern low pressure weakening under SWP4 and Japan low pressure appearance under SWP5. To clarify the above findings, we add the following explanations on lines 463, 499 and 515 in the new revised manuscript.

Line 463:

At 500 hPa, a trough located at approximate 120°E–125°E is strengthened associated with Aleutian low shifting southward,

Line 499:

At 500 hPa, a shallow trough located at about 125°E strengthens associated with weakening of the southern cyclone pressure,

Lines 515:

At 500 hPa, a trough located at about 130°E controlling the YRD strengthens associated the Japan low pressure appearance.

In comparison with the similar previous studies (Han, et al., 2020; and Gao et al., 2020), this paper is not clear in spatial distribution of pressure and lack of clear pictures in synoptics.

Thanks for your comments. In order to show clear pictures in synoptic, specific figures of atmospheric circulation at 850 hPa under each SWP are added in the supplement. As shown in Fig. 1, SWP1 is under control of the southwesterly flow introduced by the WPSH. SWP2 is influenced by the northwesterly flow introduced by a continental high pressure and the Aleutian low pressure. SWP4 is influenced by the southeasterly flow introduced by the WPSH and a cyclone. SWP3 and SWP5 are affected by a cyclone and an anticyclone. The above findings are added on lines 376-379 in the new revised manuscript as well.

These figures are similar with figures in Pos phase or Neg phase under each SWP, and we primarily explore the changes in atmospheric circulation between Pos and Neg phase of the SWPs. Therefore, these figures are only added in the supplement.

[Figure]

**Fig. 1. The geopotential height (shaded) and 850 hPa wind with temperature (color vector) under (a) SWP1, (b) SWP2, (c) SWP3, (d) SWP4, (e) SWP5. The boxed area in Figs.1a-e encloses the YRD.**

3.In figure 1, "43.40" need mention in the context.

Thanks for your comments. "43.40 ppb" represents the highest monthly mean $O_3$ concentration value during the warm seasons in 2014-2018. Fig. 2a primarily shows the increasing trend during this period, so it is inappropriate to illustrate this maximum number in the figure. In the new revised manuscript, "43.40" is deleted.

[Figure]

**Fig. 2. (a) Anomalies of monthly average O3 concentration from April to September during 2014–2018. The purple solid line represents the linear fitted curve, and the color number represents the annual (April–September) mean of O3 concentration.**

4.In the EOF analysis, the spatial distributions of EOF1 are generally negative and time series of EOF1 presents a decreasing trend. Actually, the O3 generally increase all over YRD in recent years. So, I suggest multiply -1 with spatial distributions of EOF1 and time series of EOF1 make the statement easy to follow.

Thanks for your suggestions. The spatial distributions and time series of EOF1 mode have been multiplied -1 in Fig. 3. In addition, we correspondingly change Fig. 4 and other illustrations. Please see Figs.2 and 9, and the words on lines 398-400 in the new revised manuscript. The revisions are listed as below:

[Figure]

**Fig. 3. First EOF patterns of O₃ concentration in the warm seasons from 2014 to 2018, including the spatial pattern (a) and time coefficient (b). The percentage in panels (a) is the variance contribution of each EOF mode. The pink dash line in panels (b) represents the linear fitted curve.**

[Figure]

**Fig. 4. The trend of the inter-annual EOF1 time series in the warm seasons. The pink curve represents the original inter-annual EOF1 time series in the warm seasons, the green line represents the reconstructed EOF1 time series only accounting the frequency variation in SWPs, and blue line represents the reconstructed one accounting both the frequency and the intensity variations in SWPs.**

Lines 398-400:

the positive phase (Pos) represents that the EOF1 time series is more than 0 and it is beneficial to the production and accumulation of $O_3$. On the contrary, the negative phase (Neg) corresponds low $O_3$ concentration.

5.The authors reveal RH is the key factor dominating inter-annual variations of ozone, and indicate that its unclear in the relations between RH and ozone in previous study. I suggest the authors gives a further clear explanation of the RH effects on ozone. RH could related to the cloud cover (solar radiation), stable of air in BL and so on. In figure 3b, sunshine duration (may related to cloud cover?) is not important in ozone inter-annual variation, and opposite to RH, which may implicate that stable of air (accumulation of air pollutants) is important?

Thanks for your comments. We re-quantify the meteorological factors impact on the $O_3$ variation, and stress influential mechanism on $O_3$ variation from dominated meteorological factors.

In section 3.2.1, we quantify the meteorological impact on the $O_3$ variation using meteorological adjustment method. In the original manuscript, we adopted sunshine duration, air temperature at 2m, wind speed at 10m and relative humidity as the input factors. According to the above suggestions, in order to clarify the effects of relative humidity (RH) on $O_3$, we replace the sunshine duration with solar radiation (SR) and add the low cloud cover (LCC). There are two reasons for selecting LCC to analysis. Firstly, low clouds are more effective at blocking out sunlight (SR) than medium and high clouds. Secondly, LCC has the higher correlation coefficient with SR than total cloud cover, medium cloud cover and high cloud cover. As shown in Fig. 5, RH is the most crucial factor and its variation is similar to the variation in the total meteorological impact. In addition, SR and LCC also play important roles and have large impacts on $O_3$ variation. RH can impact $O_3$ concentration in two ways. One is gas phase $H_2O$ reacting with $O_3$ ($O_3 + H_2O + hv = O_2 + 2OH$). The other is its influencing on clouds and thereby shielding SR. During this process, specifically, under low RH

circumstance, the reactions between water vapor and $O_3$ are inhibited. Moreover, low RH leads to less cloud cover, and thereby there is more intensive SR. Strong SR can enhance $O_3$ chemical reaction.

In a word, RH, SR and LCC all have important effects on $O_3$ variation. Among them, RH plays the most significant role in modulating the inter-annual $O_3$ variation. Low RH prevents $O_3$ to react with gas phase $H_2O$. Moreover, low RH caused by vertical downward motions results in less LCC and intensive SR, which can enhance the $O_3$ chemical reactions and lead to higher $O_3$ concentrations. The above-mentioned specific discussions have been added in section 3.2.1 in the new revised manuscript.

[Figure]

**Fig. 5. (a) 5-year trends of ambient O3 (solid black line), meteorological adjusted O3 (dashed black line), and the meteorological impact (pink line) over the YRD during 2014–2018. Periods with positive and negative meteorological impacts are shaded in red and green, respectively; red and green bars represent the O3 increases and decreases attributable to meteorological influence in each year. (b) 5-year variations in the meteorological impact of different meteorological factors (MEO), including relative humidity (RH), solar radiation (SR), air temperature (T2), wind speed (WS) and low cloud cover (LCC).**

6.In line 402, "the cloud cover hard to form" should be "the cloud hard to form."

Thanks for your suggestions. "the cloud cover hard to form" is changed to be "hinder cloud formation". Please see line 440 in the new revised manuscript

7.What is the unit in figure 4 of W (vertical velocity), m/s or Pa/s?

The unit of W (vertical velocity) is "Pa/s". In the new revised manuscript, the sub-figures (e) in Fig. 4, 5, 6, 7, 8 have been replaced by Table 3, and the unit of W "Pa/s" have been added in the Table 3. Please see line 527 in the new revised manuscript.

8.In line 427-428, the sentence "At 500 hPa . . .." should indicate the area of downward motion.

Thanks for your suggestions. As shown in Fig. 6a and b, the northwest YRD area in red box is located behind the strengthening trough. According to the potential tendency equation, downward motions are usually located behind the trough. Thus, in this case, the northwest YRD area is

associated with stronger downward motion. The above discussion is added in the new revised manuscript. Please see lines 462-464.

[Figure]

**Fig. 6. The geopotential height (shaded) and 500 hPa wind with temperature (color vector) under (c) SWP2_Pos and (d) SWP2_Neg. The red values represent regional average wind speed at 500 hPa in the zone around black lines. The boxed area in Figs.6a-d encloses the YRD.**

9.What is SR in fig. 4 etc.? SD?

Thanks for your suggestions. SR and SD represent the solar radiation and sunshine duration, respectively. Solar radiation reanalysis data and sunshine observation data are acquired from the ERA-interim dataset and the air quality real-time publishing platform. SR is usually regarded as the directly influential factor of $O_3$ formation. Therefore, in the section 3.2.1, we quantify the effect of meteorological conditions by using SR. In the new revised manuscript, SD is not used any more

10.From figure 4-8, a summary table with values of meteorological factors in 5 SWPs could be better than sub-figure (e) for comparisons.

Thanks for your suggestions. We change sub-figures (e) in Fig. 4, 5, 6, 7, 8 into a summary table. Table 1 shows regional mean ± the standard error of meteorological factors in Pos phase and Neg phase and their difference (Pos minus Neg) under each pattern. The meteorological factors include relative humidity (RH), solar radiation (SR), air temperature (T2), low cloud cover (LCC), total cloud liquid water (TCLW), zonal wind speed at 850 hPa (V850) and vertical motion (W).

RH, SR and T2 are dominated meteorological factors affecting $O_3$ variation. V850 is an important element of bringing water vapor to the YRD and result in RH variations. Moreover, under the condition of vertical upward or downward motion (W), RH would change low cloud cover (LCC) and total cloud liquid water (TCLW), leading to the variation of SR.

**TABLE 1. Regional mean ± the standard error of meteorological factors in Pos phase and Neg phase and their difference under each pattern.**

| SWP | phase | RH | SR(W/m²) | T2(℃) | LCC | TCLW | V850(m/s) | W(Pa/s) |
|-----|-------|------|-----------|--------|------|------|-----------|---------|
| P1 | Pos | 69.70±9.69 | 1970.97±403.19 | 29.90±4.76 | 0.07±0.15 | 0.06±0.08 | 2.89±2.24 | 0.00±0.05 |

| | | | | | | | |
|---|---|---|---|---|---|---|---|
| | Neg | 84.94±6.53 | 1240.93±460.18 | 27.45±4.78 | 0.37±0.27 | 0.17±0.14 | 4.27±2.73 | -0.05±0.05 |
| | Diff | -15.24 | 730.04 | 2.45 | -0.30 | -0.11 | -1.38 | 0.05 |
| | Pos | 66.49±10.96 | 1968.41±377.12 | 28.81±4.32 | 0.07±0.14 | 0.06±0.09 | -2.47±3.09 | 0.02±0.05 |
| P2 | Neg | 81.29±10.78 | 1178.34±479.58 | 23.89±5.90 | 0.48±0.31 | 0.19±0.14 | -1.37±3.21 | -0.03±0.06 |
| | Diff | -14.79 | 790.06 | 4.91 | -0.41 | -0.13 | -1.10 | 0.05 |
| | Pos | 76.89±7.09 | 1371.42±605.82 | 27.83±2.45 | 0.34±0.18 | 0.21±0.19 | -0.67±3.43 | -0.02±0.04 |
| P3 | Neg | 88.62±5.14 | 854.96±395.09 | 24.77±4.58 | 0.58±0.24 | 0.31±0.16 | 1.93±3.65 | -0.09±0.06 |
| | Diff | -11.73 | 516.45 | 3.06 | -0.24 | -0.10 | -2.60 | 0.07 |
| | Pos | 71.11±7.15 | 1882.33±388.10 | 30.62±3.69 | 0.11±0.16 | 0.12±0.16 | 0.57±2.40 | 0.01±0.04 |
| P4 | Neg | 83.37±6.76 | 1343.80±547.50 | 28.93±4.19 | 0.35±0.24 | 0.19±0.19 | 2.46±3.60 | -0.04±0.06 |
| | Diff | -12.26 | 538.53 | 1.69 | -0.24 | -0.07 | -1.89 | 0.05 |
| | Pos | 68.47±14.19 | 1827.46±447.37 | 29.60±5.25 | 0.07±0.11 | 0.09±0.14 | -1.83±3.42 | 0.01±0.04 |
| P5 | Neg | 85.81±3.45 | 1199.21±397.17 | 26.43±3.82 | 0.43±0.30 | 0.16±0.09 | -2.31±5.25 | -0.02±0.04 |
| | Diff | -17.34 | 628.26 | 3.17 | -0.35 | -0.07 | 0.48 | 0.03 |
| Others | | / | / | | / | | / | / |

11.In section 3.4, I wonder why do you reconstruct the EOF1 time series? It could be more valuable to reconstruct the inter-annual variations of ozone concentration based on SWPs frequencies and intensities. And What's SWPIIs?

Thanks for your suggestions. We reconstruct the EOF1 time series to replace the regional mean $O_3$ concentration. There are two reasons for this decision. Firstly, the time series of EOF1 shows a high negative correlation with the $O_3$ time series (R = -0.98). More importantly, we primarily focus on why $O_3$ concentration increases in the entire YRD region, rather than why the increases in $O_3$ differ spatially inside the YRD. Therefore, it is more appropriate to reconstruct EOF1 time series than $O_3$ time series.

SWPIIs represent synoptic weather pattern intensity indexes. They are defined as maximum geopotential height in zone 1(25°N–40°N, 110°E–130°E) for SWP3 and SWP5, maximum geopotential height in zone 2 (20°N–50°N, 90°E–140°E) for SWP1 and SWP4, and average geopotential height in zone 3 (10°N –40°N, 110°E–130°E) for SWP2, according to their high correlation coefficients with EOF1 time series under each SWP. Especially, zone1, 2 and 3 were selected in term of location of dominated weather systems under each SWP. Please see lines 255-257 and 559-570 in the new revised manuscript.

Technical corrections: 1.In the caption of figure 2, are they "orange dash line"? Pink? 2.There are several typo need carefully check. For example, meddle in line 313; "wins" in line 468; "SR" in 336 could be SD.

Thanks for your suggestions. The above-mentioned typos have been corrected. For example, "orange", "meddle" and "wins" are modified as "pink", "middle", "winds". Please see lines 314, 332 and 495 in the new revised manuscript. In the new revised manuscript, Sunshine duration (SD) is not used any more, as mentioned in the response to the specific comment 9.

**Response to the comments of Referee #2:**

This paper discussed the meteorological influence on the increase in ozone concentration in the YRD china. Abundant analysis methods were used to try to figure out the reason to the increase in the ozone concentration. The results are some helpful to ozone pollution control and prediction. I have some comments in the following to improve this paper.

Thanks for your comments and suggestions. All your comments and suggestions are very important. They have directive significance to our paper writing and research work.

General comments: You attributed the effect of low RH on the increase in the ozone concentration to the strong solar radiation and high temperature in all kinds of SWPs. However, why does RH show a significant correlation with ozone concentration instead of the more direct meteorological factors temperature and radiation? Moreover, as you said, the lack of clouds contribute much to the high concentration of ozone. Why not analyze the impact of cloud property (such as cloud fraction, cloud thickness, cloud height, cloud liquid content) in the meteorological dataset? Cloud is the direct impact factor probably. Through the results and discussion section, they are almost qualitative description. Additional quantitative analysis and discussion are needed to make your conclusion more significant and scientific. The discussions in S3.3.2 about the impacts of SWP on ozone concentration are too similar for five SWPs. They all results in the downward motion, high temperature, strong radiation. I suggest to pay more attention to the difference in the impact among SWPs.

Thanks for your comments. Combining this comment and the referee1's comments, we re-quantify the effects of meteorological factors on $O_3$ variation. The factors include relative humidity (RH), solar radiation (SR), air temperature (T2), wind speed (WS) and low cloud cover (LCC). We replace sunshine duration (SD) by SR, and add the new factor LCC. There are two reasons for selecting LCC to analysis. Firstly, low clouds are more effective at blocking out sunlight (SR) than medium and high clouds. Secondly, LCC has the higher correlation coefficient with SR than total cloud cover, medium cloud cover and high cloud cover.

In this study, SR has a significant positive correlation with ozone concentration (R = 0.56), and there is also a high negative correlation coefficient between RH and $O_3$ (R = -0.59). As shown in Fig. 1b, RH is the most crucial factor and its variation is similar to the variation in the total meteorological impact. In addition, SR and LCC also play important roles and have large impacts on $O_3$ variation. RH can impact $O_3$ concentration in two ways. One is gas phase $H_2O$ reacting with $O_3$ ($O_3 + H_2O + hv = O_2 + 2OH$). The other is its influencing on clouds and thereby shielding SR. Under low RH circumstance, the reactions between gas phase $H_2O$ and $O_3$ are inhibited. Moreover, low RH leads to less LCC, and thereby there is more intensive SR. Stronger SR can enhance $O_3$ chemical reaction.

LCC also play important role and have large impacting on $O_3$ variation. Beside the shielding effect of clouds, total column cloud liquid water (TCLW) can influence reflection above the cloud. Therefore, TCC and TCLW are both considered to indicate the effect of RH on SR in section 3.3.2. As shown in table 2, as the radiation (SR) increases, LCC and TCLW present different extent of decreasing under each SWP. The corresponding explanations are added in section 3.2.1

To make conclusion more significant and scientific, and to reveal the different impact of each SWP on ozone concentration, we quantitatively analysis the difference in meteorological factors between Pos phase and Neg phase under each SWP. Table 2 shows the decreasing of RH, LCC, TCLW and V850 (meridional wind at 850 hPa) and the increasing of SR, T2 and W (vertical velocity) under all SWPs. It indicates that the decreasing of RH leads to the decreasing of LCC and TCLW under the condition of vertical downward motion, and thereby causes the strengthening of SR. However, the decreasing and the increasing of meteorological factors are obviously different under each pattern. Therefore, crucial meteorological factors leading to increases in O$_3$ concentrations are different under different SWPs. We calculate the correlation coefficients between the EOF1 time series and these meteorological factors (such as RH, SR and T2) under each SWP. As shown in Table 1 and 2, when the absolute values of the calculated correlation coefficients under a SWP are greater than 0.4, the corresponding meteorological factors present significant changes between Pos and Neg phases. Therefore, we regard them as the crucial meteorological factors that impact O$_3$ variation under that SWP. In the end, we find that significant decreases in RH and increases in SR are the crucial meteorological factors under SWP1, SWP4 and SWP5. For SWP2, significant decreases in RH, increases in SR and T are the crucial meteorological factors. For SWP3, significant decreases in RH is the crucial meteorological factor.

In summary, quantitatively analyses are added through substantial meteorological factors comparison between Pos phase and Neg phase. In addition, we explore correlation coefficients of dominated meteorological factors with EOF1 time series under each SWP. Combining high correlation coefficients and big difference of dominated meteorological factors in two phase, we find that crucial meteorological factors impacting on O$_3$ variation are different under each SWP. Please see the added specific discussion in section 3.3.2 in the new revised manuscript.

[Figure]

**Fig. 1. (a) 5-year trends of ambient O3 (solid black line), meteorological adjusted O3 (dashed black line), and the meteorological impact (pink line) over the YRD during 2014–2018. Periods with positive and negative meteorological impacts are shaded in red and green, respectively; red and green bars represent the O3 increases and decreases attributable to meteorological influences in each year. (b) 5-year variations in the meteorological impact of different meteorological factors (MEO), including relative humidity (RH), solar radiation (SR), air temperature (T2), wind speed (WS) and low cloud cover (LCC).**

**TABLE 1. Correlation coefficients of RH, SR and T2 with EOF1 time series under each SWP.**

| Vars | SWP1 | SWP2 | SWP3 | SWP4 | SWP5 |
|------|------|------|------|------|------|
| RH | 0.59 | 0.52 | 0.50 | 0.64 | 0.59 |
| SR | -0.58 | -0.56 | -0.33 | -0.46 | -0.48 |
| T2 | -0.19 | -0.41 | -0.26 | -0.15 | -0.30 |

**TABLE 2. regional mean $\pm$ the standard error of meteorological factors in Pos phase and Neg phase and their difference under each pattern.**

| SWP | phase | RH(%) | SR(W/m$^2$) | T2(℃) | LCC | TCLW | V850(m/s) | W(Pa/s) |
|-----|-------|-------|-------------|-------|-----|------|-----------|---------|
| | Pos | 69.70±9.69 | 1970.97±403.19 | 29.90±4.76 | 0.07±0.15 | 0.06±0.08 | 2.89±2.24 | 0.00±0.05 |
| P1 | Neg | 84.94±6.53 | 1240.93±460.18 | 27.45±4.78 | 0.37±0.27 | 0.17±0.14 | 4.27±2.73 | -0.05±0.05 |
| | Diff | -15.24 | 730.04 | 2.45 | -0.30 | -0.11 | -1.38 | 0.05 |
| | Pos | 66.49±10.96 | 1968.41±377.12 | 28.81±4.32 | 0.07±0.14 | 0.06±0.09 | -2.47±3.09 | 0.02±0.05 |
| P2 | Neg | 81.29±10.78 | 1178.34±479.58 | 23.89±5.90 | 0.48±0.31 | 0.19±0.14 | -1.37±3.21 | -0.03±0.06 |
| | Diff | -14.79 | 790.06 | 4.91 | -0.41 | -0.13 | -1.10 | 0.05 |
| | Pos | 76.89±7.09 | 1371.42±605.82 | 27.83±2.45 | 0.34±0.18 | 0.21±0.19 | -0.67±3.43 | -0.02±0.04 |
| P3 | Neg | 88.62±5.14 | 854.96±395.09 | 24.77±4.58 | 0.58±0.24 | 0.31±0.16 | 1.93±3.65 | -0.09±0.06 |
| | Diff | -11.73 | 516.45 | 3.06 | -0.24 | -0.10 | -2.60 | 0.07 |
| | Pos | 71.11±7.15 | 1882.33±388.10 | 30.62±3.69 | 0.11±0.16 | 0.12±0.16 | 0.57±2.40 | 0.01±0.04 |
| P4 | Neg | 83.37±6.76 | 1343.80±547.50 | 28.93±4.19 | 0.35±0.24 | 0.19±0.19 | 2.46±3.60 | -0.04±0.06 |
| | Diff | -12.26 | 538.53 | 1.69 | -0.24 | -0.07 | -1.89 | 0.05 |
| | Pos | 68.47±14.19 | 1827.46±447.37 | 29.60±5.25 | 0.07±0.11 | 0.09±0.14 | -1.83±3.42 | 0.01±0.04 |
| P5 | Neg | 85.81±3.45 | 1199.21±397.17 | 26.43±3.82 | 0.43±0.30 | 0.16±0.09 | -2.31±5.25 | -0.02±0.04 |
| | Diff | -17.34 | 628.26 | 3.17 | -0.35 | -0.07 | 0.48 | 0.03 |
| Others | / | / | / | | / | | / | / |

Line 115: How many sites in total in 26 cities were used in your research? Or you used the mean concentration for each city?

Thanks for your comments. There are total 172 stations in 26 cities. The data were acquired from the air quality real-time publishing platform (http://106.37.208.233:20035/) and the National Meteorological Center of China Meteorological Administration. In order to better characterize the $O_3$ pollution levels of each city, the hourly $O_3$ concentration of each city is calculated as the average value of the $O_3$ concentrations measured in several of the national monitoring sites in that city. Please see lines 123-125 in the new revised manuscript.

Line 125: How many missing data in your dataset? Can you evaluate the influence of these missing data on your conclusion?

Thanks for your comments. There are 1487 missing data in total 21960 $O_3$ data, accounting for 6.77%. By offsetting the missing data with random number range from the minimum to maximum at that time, we conduct the two random experiment in comparison to the inconsiderable missing one, named random1, random2 and original experiment, respectively. Regarding the main conclusion, the inter-annual $O_3$ trend, EOF modes, meteorological adjustment results are more

easily affected by missing data than average change in atmospheric circulation and meteorological factors and reconstructed yearly EOF1 time series. Therefore, we primarily contrast the first three results. In addition, EOF modes were obtained by deleting missing data of 17 days due to the requirement of algorithm and we just offset 1079 (1487-17*24) missing data like before. The results are as following. In Fig. 2, inter-annual increasing trends are 5.21%, 5.28% and 5.13% in original, random1 and random2 experiments. In Fig. 3, the maximum meteorological contributions are 3.03, 3.22 and 3.00 ppb in original, random1 and random2 experiments. In Fig. 4, variation contributions of first EOF mode are 65.7%, 65.6% and 65.6% and the correlation coefficients of EOF1 time series in original experiment with other two random experiment are 0.99 and 0.99. The above results in original experiment are similar with those in other two random experiments. So the missing data has little influence to our conclusions.

[Figure]

**Fig. 2. Anomalies of monthly average O₃ concentration from April to September during 2014–2018 in original (a), random1 (b) and random2 (c) experiments. The purple solid line represents the linear fitted curve, and the color number represents the annual (April–September) mean of O₃ concentration.**

[Figure]

**Fig. 3.** 5-year trends of ambient O₃ (solid black line), meteorological adjusted O₃ (dashed black line), and the meteorological impact (pink line) over the YRD during 2014–2018 in original (a), random1 (b) and random2 (c) experiments. Periods with positive and negative meteorological impacts are shaded in red and green, respectively; red and green bars represent the O₃ increases and decreases attributable to meteorological influences in each year.

[Figure]

**Fig. 4.** First EOF patterns of O₃ concentration in the warm seasons from 2014 to 2018 in original (a), random1 (b) and random2 (c) experiments, including the spatial pattern. The percentage in panels (a), (b) and (c) are the variance contribution of each EOF mode.

Line 130: Please list the number of coefficients you used in this function.

Thanks for your comments. As the formulate shown in equation (1) in the new revised manuscript and Fig. 5a, the 1.81 of $k$ value is the linear trend, regarded as the inter-annual $O_3$ variation trend during the warm season in 2014-2018. It is used in this function and as a conclusion in this study. In order to make readers seize the used coefficient number, corresponding explanation is added in lines 144-145 in the new revised manuscript.

[Figure]

Fig. 5. (a) Anomalies of monthly average $O_3$ concentration from April to September during 2014–2018. The purple solid line represents the linear fitted curve, and the color number represents the annual (April–September) mean of $O_3$ concentration.

Lines 144-145:

In this study, linear trend $k$ is regarded as the inter-annual $O_3$ variation trend and is discussed in section 3.1.1.

Line 244: I suggest to use all data instead of monthly mean over 26 cities to do linear fitting because some extreme high concentration in several cities may change the fitting results. Please show the fitting function and correlation coefficient.

Thanks for your comments. In this section, we want to archive the inter-annual $O_3$ variation. Hourly $O_3$ data contain too many temporal variation signals such as $O_3$ hour-to-hour variation, day-to-day variation and seasonal variation. If we carry out the linear regression using the hourly $O_3$ data, the $k$ value of fitting curve cannot be regarded as the inter-annual $O_3$ variation. In addition, the linear regression using fewer $O_3$ year-to-year data are easily overfitting. Therefore, monthly mean $O_3$ concentrations are adopt. The inter-annual $O_3$ variation can be obtained through separating the seasonal signal in the linear trend model. We added above discussion in lines 134-135 of the new revised manuscript.
The fitting function and corresponding explanation is also added in section 3.1.1. Please see lines 268-269 in the new revised manuscript.

Line 272: How did you define the coefficient of meteorological factors like WPSH, EASM? How

did you calculate the correlation between meteorological factors and ozone concentration? Fig3: The abbreviations are different in the figure and captions.

Thanks for your comments.
The WPSH index (WI) is defined according to WPSH intensity index in the National Climate Center of China. It is characterized by the sum of the product of the total area encircled by the 5880 gpm isolines within the range of 110°E–180°E and north of 180°N, and the difference between the grid point's gpm and 5870 gpm.
The EASMI is a shear vorticity index. It is defined as the difference of the regional mean zonal wind at 850 hPa between 5 and 15°N, 22.5 and 32.5°N, 90 and 130°E, and 110 and 140°E in Wang and Fan (1999), recommended by Wang et al. (2008).
We added the WPSHI and EASMI definitions in section 2.5. Please see lines 216-223 in the new revised manuscript.

References
Wang, B., and Fan, Z.: Choice of south Asian summer monsoon indices, B Am Meteorol Soc, 80, 629-638, Doi 10.1175/1520-0477(1999)080<0629:Cosasm>2.0.Co;2, 1999.
Wang, B., Wu, Z. W., Li, J. P., Liu, J., Chang, C. P., Ding, Y. H., and Wu, G. X.: How to measure the strength of the East Asian summer monsoon, J Climate, 21, 4449-4463, 10.1175/2008JCLI2183.1, 2008.

We calculate the correlations between them (WPSHI and EASMI) and ozone concentration according to the Pearson Correlation coefficient. The correlation is significant at 0.01 confidence level.
Pearson correlation coefficient as the calculating correlation coefficient method has been added in lines 227-228 of the new revised manuscript. Besides, Pearson correlation coefficient is widely known to all, it is unnecessary to introduce its calculation formula in the new revised manuscript.

In Figure 3, "SD" in (b), and "MER" and "WP" in captions have been replaced by "SR", "MEO" and "WS". Please see line 350 and lines 356-357 in the new revised manuscript.

Line 352: Here you said "SWP1 is affected by the southeasterly flow. . .", while "Southwesterly flow" for SWP1 in the Table 1.

Thanks for your comments. "southeasterly" has been corrected into "southwesterly" in the new manuscript. Please see line 376 in the new revised manuscript.

Section 3.3.1: It is better to show these six SWPs in figure addition to the Table 1, at least in the supplementary.

Thanks for your comments. In order to show clear pictures in synoptic, specific figures of atmospheric circulation at 850 hPa under each SWP are added in Fig. 6. As shown in Fig. 6, SWP1 is under control of the southwesterly flow introduced by the WPSH. SWP2 is influenced by the northwesterly flow introduced by a continental high pressure and the Aleutian Low pressure. SWP4 is influenced by the southeasterly flow introduced by the WPSH and a cyclone. SWP3 and SWP5

are affected by a cyclone and an anticyclone. The above findings are added in lines 376-379 in the new revised manuscript as well.

These figures are similar with figures in Pos phase or Neg phase under each SWP, and we primarily explore the changes in atmospheric circulation between Pos and Neg phase of the SWPs. Therefore, these figures are only added in the supplementary.

[Figure]

**Fig. 6. The geopotential height (shaded) and 850 hPa wind with temperature (color vector) under (a) SWP1, (b) SWP2, (c) SWP3, (d) SWP4, (e) SWP5. The boxed area in Figs.6a-e encloses the YRD.**

Table 1: What do the meteorological factors mean? Regional average during all warm seasons?

Meteorological factors include air temperature, relative humidity, wind speed and solar radiation etc. They have directly and indirectly impacts on the formation of $O_3$ pollution. Region mean values of the meteorological factors are calculated in each SWP, not during all warm seasons. The corresponding explanation are added in lines 386-387 of the new revised manuscript.

Line 379: How did you analyze the daily variation? What is the influence on the result?

Thanks for your comments. The daily variation is related to the day-to-day variation of SWPs. In this study, we use the positive phase (Pos) and the negative phase (Neg) to study the changes in $O_3$. As shown in Fig. 7a and b, the Pos represents that the EOF1 time series is more than 0 and it is beneficial to the production and accumulation of $O_3$. On the contrary, the Neg corresponds low $O_3$ concentrations. In Yin et al. work, changes in atmospheric circulation were obtained by comparing Pos with Neg during whole period. We also try to gain the changes in atmospheric circulation through this method. As shown in Fig 8, it can only shows the change in WPSH, and changes in weather systems would be hiding. Therefore, we first extract the predominant SWPs in the warm seasons over the YRD using a weather classification method. And then, changes in atmospheric circulations are obtained by comparing Pos with Neg under each SWP.

There are large differences on the results between our method and Yin's method. Fig. 8 and Fig. 9 show the results obtained from Yin's method and our method. In Fig. 8, only change in WPSH is clearly shown, and changes in weather systems would be hiding. But in Fig. 9, beside WPSH, changes in other weather systems including a continental high pressure and the Aleutian low pressure under SWP2, a cyclone under SWP3 and SWP4, and an anticyclone under SWP5 can also be obtained.

[Figure]

**Fig. 7. First EOF patterns of $O_3$ concentration in the warm seasons from 2014 to 2018, including the spatial pattern (a) and time coefficient (b). The percentage in panels (a) is the variance contribution of each EOF mode. The pink dash line in panels (b) represents the linear fitted curve.**

[Figure]

**Fig. 8. The geopotential height (shaded) and 850 hPa wind with temperature (color vector) under (a) Pos and (b) Neg. The boxed area in Figs.8a-b encloses the YRD.**

[Figure]

**Fig. 9. The geopotential height (shaded) and 850 hPa wind with temperature (color vector) under (a) SWP1_Pos, (b) SWP1_Neg, (c) SWP2_Pos, (d) SWP2_Neg, (e) SWP3_Pos, (f) SWP3_Neg, (g) SWP4_Pos, (h) SWP4_Neg, (i) SWP5_Pos and (j) SWP5_Neg. The boxed area in Figs.9a-j encloses the YRD.**

Line 384: How can you get the conclusion that frequency change has less impact than the intensity change? Please add quantitative evaluation.

Thanks for your comments. In the new revised manuscript, it is emphasized that we get the conclusion (the frequency change has less impact than the intensity change) based on Fig. 10 and the relevant quantitative evaluation. Fig. 10 shows the trend of the inter-annual EOF1 time series. The pink curve represents the original inter-annual EOF1 time series, the green line represents the reconstructed ones only accounting the frequency variation in SWPs, and the blue line represents the reconstructed ones accounting both the frequency and the intensity variations in SWPs. By comparing original EOF1 time series with the two reconstructed ones, we find out the importance of the intensity change and the frequency change to inter-annual $O_3$ variation. In this study, we define the contribution index as the difference between the maximum and the minimum of a certain reconstructed time series divided by the difference between the maximum and the minimum of annual EOF1 time series: Contribution Index = (The reconstructed maximum – the reconstructed minimum)/(the original maximum – the original minimum). Through the above equation, we derive the relative contribution (contribution index) of the frequency change and the intensity change. Compared with the contribution index of 10.86% for SWPs frequency change, the value of 48.89% for SWPs intensity change accounts for a larger proportion. Therefore, the intensity change in SWP is more important to the inter-annual $O_3$ variation than the frequency change.

[Figure]

**Fig. 10. The trend of the inter-annual EOF1 time series in the warm seasons. The pink curve represents the original inter-annual EOF1 time series in the warm seasons, the green line represents the reconstructed EOF1 time series only accounting the frequency variation in SWPs, and blue line represents the reconstructed one accounting both the frequency and the intensity variations in SWPs.**

Line 393: Please describe the difference in WPSH using some representative index like WPSH index, ridge position, instead of the puzzled word "wider".

Thanks for your comments. We stress the difference between each SWP in Section 3.3.2 of the new revised manuscript. Thus, we do not specially emphasis the changes in WPSH area under SWP1, and delete the relevant words. In addition, we point out the position of ridge under each SWP. Please see lines 438, 462, 481, 499 and 515 in the new revised manuscript.

Line 399: If the downward air mass comes from ocean with abundant water vapor, although the cloud is hard to form, the RH on the surface possibly increases. How can you explain the negative correlation between surface RH and ozone concentration? same question for the explanation of other SWPs.

Thanks for your comments. According to the Table 2, even if the downward motion strengthens, RH still shows a decreasing trend under each SWP due to the weakening transportation of air masses from the southern and eastern sea areas.

For the negative correlation between surface RH and ozone concentration, it is related to two fact. Firstly, under low RH circumstance, the reactions between gas phase $H_2O$ and $O_3$ are inhibited. Secondly, low RH leads to less LCC, and thereby there is more intensive SR. Stronger SR can enhance $O_3$ chemical reaction.

Line 511: I don't think it is obvious that frequency changes have on impact. It looks that the contribution from frequency changes is comparable to that from intense changes according to Fig9. Could you give more explanation or evidence?

Thanks for your comments. In order to accurately evaluate the contribution to $O_3$ variation from SWP frequency change and intensity change, quantitative evaluation is added in the new revised manuscript.

Fig. 10 shows the trend of the inter-annual EOF1 time series. The pink curve represents the original inter-annual EOF1 time series, the green line represents the reconstructed ones only accounting the frequency variation in SWPs, and the blue line represents the reconstructed ones accounting both the frequency and the intensity variations in SWPs. By comparing original EOF1 time series with the two reconstructed ones, we find out the importance of the intensity change and the frequency change to inter-annual $O_3$ variation. In this study, we define the contribution index as the difference between the maximum and the minimum of a certain reconstructed time series divided by the difference between the maximum and the minimum of annual EOF1 time series: Contribution Index = (The reconstructed maximum – the reconstructed minimum)/(the original maximum – the original minimum). Through the above equation, we derive the relative contribution (contribution index) of the frequency change and the intensity change. Compared with the contribution index of 10.86% for SWPs frequency change, the value of 48.89% for SWPs intensity change accounts for a larger proportion. Therefore, the intensity change in SWP is more important to the inter-annual $O_3$ variation than the frequency change. We add the above discussion on lines 539-547 of the new revised manuscript.

Line 517: What are patter V?

Thanks for your comments. The "Pattern V" is described in the previous study of Hegarty et al. (2007), where Hegarty et al. reconstructed the inter-annual $O_3$ pollution level in the northeastern United States using the similar method as ours. They define the intensity change index in SWPs using the domain-averaged sea level pressure. In this study, we observed the poor correlation between the $O_3$ pollution level and the intensity change under "pattern V". Therefore, in order to optimize SWP intensity index, we define the SWPIIs under each pattern according to their unique characteristics response to high $O_3$ concentration. In table 4, the correlations under each pattern have been improved. In order to avoid confusing, we replace "Pattern V" by "Hegarty's Pattern V". Please see line 553 in the new revised manuscript.

Line 517: What is the definition of "SWPII"? How did you calculate it? It is better to show the number for each SWP.

Thanks for your comments. SWPIIs represent synoptic weather pattern intensity indexes. They are defined as maximum height in zone 1(25°N–40°N, 110°E–130°E) for SWP3 and SWP5, maximum height in zone 2 (20°N–50°N, 90°E–140°E) for SWP1 and SWP4, and average height in zone 3 (10°N –40°N, 110°E–130°E) for SWP2, according to their high correlation coefficients with EOF1 time series under each pattern. Especially, zone1, 2 and 3 were selected in term of location of dominated weather systems under each SWP. Please see lines 255-256 and 539-547 in the new revised manuscript

For the number of SWPII for each SWP, we present them in Table 3, which is added in the supplement because the numbers have no direct relations to our conclusion.

**TABLE 3. SWPIIs under each pattern (unit: gpm).**

| Year | 2014 | 2015 | 2016 | 2017 | 2018 |
|------|------|------|------|------|------|
| SWPII1 | 1541.19 | 1547.36 | 1551.93 | 1551.12 | 1548.86 |
| SWPII2 | 1432.79 | 1437.07 | 1424.71 | 1444.86 | 1443.86 |
| SWPII3 | 1520.25 | 1514.59 | 1526.00 | 1513.78 | 1519.62 |
| SWPII4 | 1546.26 | 1554.17 | 1547.95 | 1537.75 | 1551.48 |
| SWPII5 | 1517.13 | 1522.64 | 1524.00 | 1513.50 | 1512.72 |

Line 551: I did not find much quantitatively analysis in your discussion, but it should be needed.

Thanks for your comments. In order to make conclusion more significant and scientific, we quantitatively analysis meteorological factor differences between Pos phase and Neg phase under each SWP. In addition, we calculate correlation coefficients of dominated and directed meteorological factors including RH, SR and T2 with EOF1 time series under each SWP and find the main difference under them. Comprehensive considerations are as following.

Table 2 shows the decreasing of RH, LCC, TCLW and V850 and the increasing of SR, T2 and W

under all SWPs. It indicates that the decreasing of RH leads to the decreasing of LCC and TCLW under the condition of vertical downward motion, and thereby causes the strengthening of SR. However, the decreasing and the increasing of meteorological factors are obviously different under each pattern. Therefore, crucial meteorological factors leading to increases in $O_3$ concentrations are different under different SWPs. We calculate the correlation coefficients between the EOF1 time series and these meteorological factors (such as RH, SR and T2) under each SWP. As shown in Table 1 and 2, when the absolute values of the calculated correlation coefficients under a SWP are greater than 0.4, the corresponding meteorological factors present significant changes between Pos and Neg phases. Therefore, we regard them as the crucial meteorological factors that impact $O_3$ variation under that SWP. In the end, we find that significant decreases in RH and increases in SR are the crucial meteorological factors under SWP1, SWP4 and SWP5. For SWP2, significant decreases in RH, increases in SR and T are the crucial meteorological factors. For SWP3, significant decreases in RH is the crucial meteorological factor.

In section 3.3.2, it is discussed how to lead to crucial meteorological factors variation induced by change in atmospheric circulation. Please see specific discussion in section 3.3.2

---

## Author Response (AR2)

**Response to the comments of editor:**

Please check significant digits for all numbers. For example, 65.70% in line 23, 3.03 ppb in

line 25, and all values through to 1970.97 W/m2 for SR in Table 3 etc are to be checked

carefully as if their last digits are meaningful, considering large uncertainty ranges.

Thanks for your comments. All values have been checked carefully. Some values have been corrected, and others sustain original significant digits. For all changed or unchanged numbers, we give the corresponding reasons. Specific illustrations are listed as follows.

Firstly, we introduce the data we used. R\_table1 lists the number of significant digits for raw data of all variables. Secondly, we introduce the significant digits rules when we calculate mean values of all variables.

| Variables   | O 3 | T2 | RH | WS | SR | LCC | TCLW | V850 | W |
|-------------|-----------------------|----|----|----|----|-----|------|------|---|
| Number of   | 2                     | 3  | 2  | 2  | 7  | 7   | 7    | 7    | 7 |
| significant |                       |    |    |    |    |     |      |      |   |
| digits      |                       |    |    |    |    |     |      |      |   |

R table1. Significant digits of raw data of all variables

T2, RH, WS, SR, LCC, TCLW, V850 and W represent air temperature, relative humidity, wind speed at surface layer, solar radiation, low cloud cover, total liquid cloud water, zonal wind speed at 850 hPa and vertical speed, respectively.

**The significant digits rules**

We know how to get the significant digit of one value (for example, the significant digits of 20 are 2). But when we calculate the mean values, the total number of values we used has to be taken into consideration. If we calculate the daily mean value, the significant digits of the sum and mean value of this array are 3. Example

Sum value: 20×24= 480

Mean value: 480/24= 20.0

In this paper, before  $O_3$  and other meteorological factors are analyzed, we calculated their daily mean values, According to the significant digits rules and R\_table1, for all variables, the tenths place of the average is a significant digit, we can retain the tenths place at least. Therefore, we changed all values except the fitting function in section 3.1.1, percentages of variance contribution in Section 3.1.2,  $O_3$  variations in Section 3.2.1, all values in Table 1, all values in Table 3 except LCC, TCLW and W, and SWPs frequency and intensity contribution index in Section 3.4. These values are retained one decimal place.

Besides, in order to distinguish some closed numbers, all correlation coefficients and LCC, TCLW and W in table 3 retain two decimal places. And for the purpose of refined

calculation, fitting function in section 3.1.1 retains three decimal places.

Please see all corrections in the new revised manuscript with modification marks in appendix.

**Appendix**

Ozone Variability Induced by Synoptic Weather Patterns in Warm Seasons of 2014–2018 over the Yangtze River Delta Region, China

Da Gao1, Min Xie1\*, Jane Liu2,3, Tijian Wang1, Chaoqun Ma1,a, Haokun Bai1, Xing Chen1, Mengmeng Li1, Bingliang Zhuang1, Shu Li1

[revised manuscript text omitted]